# LoKiFormer: Locality-aware Attention with Decoupled Knowledge Memory for Efficient Large Language Model Pretraining

**Qiuwu Chen** [* 1] **Zimo Liu** [* 1] **Yuchen Li** [1] **Ying Sun** [1] **Yifan Zhang** [1] **Zhijie Qiu** [2] **Zeng You** [2] **Ryan Dong** [1] **Simeng Ma** [1] **Yaofo Chen** [2 3] **Mingkui Tan** [2]

## Abstract

Large language models (LLMs) have achieved remarkable breakthroughs across various applications. However, their architectures remain inefficient in pretraining due to two main limitations: (i) self-attention lacks an explicit inductive bias for locality, leading to redundant modeling of sequence-internal local information; (ii) mixture-of-experts (MoE) implicitly couples knowledge storage with computational pathways, hindering flexible access to sequence-external global knowledge. To overcome these limitations, we propose LoKiFormer, a novel LLM architecture that augments the standard decoder with two dedicated modules: 1) Local Fusion Attention (LFA), which incorporates a convolutional fusion to attention, explicitly capturing local patterns and allowing the attention to operate on more informative representations; 2) Knowledge Memory Module (KMM), which introduces a parametric key–value memory that explicitly stores global knowledge in addressable slots, decoupling storage from computation and enabling direct knowledge retrieval. Together, these modules enable LoKiFormer to achieve more efficient and effective integration of information at both levels. Experimental results show that LoKiFormer converges $1.33\times$ faster in pre-training than baseline models, underscoring its superiority over existing LLM architectures.

## 1. Introduction

Large language models (LLMs) (OpenAI, 2023; Touvron et al., 2023; Guo et al., 2025) have achieved remarkable

---
[*]Equal contribution  [1]AIGCode  [2]South China University of Technology  [3]Pazhou Laboratory. Correspondence to: Mingkui Tan <mingkuitan@scut.edu.cn>, Yaofo Chen <chenyaofo@scut.edu.cn>.

*Proceedings of the $43^{rd}$ International Conference on Machine Learning*, Seoul, South Korea. PMLR 306, 2026. Copyright 2026 by the author(s).

breakthroughs in natural language processing, demonstrating strong capabilities in applications such as customer service dialogue (Ou et al., 2024) and virtual assistants (Liu et al., 2025; Wang et al., 2025a). The core components of current LLMs primarily include 1) the attention mechanism (Vaswani et al., 2017), which models contextual dependencies by computing correlations across tokens in a sequence; and 2) the mixture of experts (MoE) (Shazeer et al., 2017), which enhances expressive power VIA expert routing and linear transformations. These components form the foundational architecture of modern LLMs, playing a central role for strong performance.

LLM architectures are expected to incorporate efficient mechanisms for modeling and processing information at different levels (Naveed et al., 2025): 1) Local-level information, which is internal to the input sequence, encompassing syntactic structures and short-range dependencies between adjacent tokens. ffectively modeling such local patterns is crucial for capturing the finer-grained semantics within a sequence, as these local structures often form the foundation for higher-level meaning extraction (Yang et al., 2021; Chen et al., 2025). 2) Global-level information, on the other hand, pertains to knowledge that is independent of any specific input sequence and includes broader concepts such as commonsense knowledge, factual information. This type of information can be internalized during pre-training and is vital for tasks that require deep reasoning and generalization beyond the scope of a single sequence (Geva et al., 2021; Mu & Lin, 2025). The challenge lies in effectively combining these local patterns with the global knowledge to empower LLMs for complex tasks.

However, we argue that existing LLM architectures are computationally inefficient during pretraining in integrating these two levels of information (See empirical analysis in Section 5.4). For modeling sequence-internal local information, while attention mechanisms can capture both short- and long-range dependencies, it does so through redundant full pairwise interactions, which makes learning local patterns computationally inefficient. For incorporating sequence-external global information, MoE implicitly distribute knowledge across the weights of expert networks.

This couples knowledge storage with computational transformation, making that knowledge retrieval becomes indirect, and the process of updating or accessing specific knowledge is inflexible. Thus, designing an architecture that efficiently and effectively integrates local and global information remains an open challenge.

In this paper, we propose LoKiFormer[1], a novel LLM architecture built upon an enhanced decoder block for efficient pretraining. It extends the vanilla decoder with two new modules: the Local Fusion Attention (LFA) and the Knowledge Memory Module (KMM), which are designed to handle sequence-internal local information and sequence-external global information. For modeling local information, LFA incorporates a convolution operation that fuses adjacent token representations before the attention mechanism. This introduces an explicit local inductive bias, allowing the model to capture short-range dependencies more efficiently in pretraining while enabling the attention layer to concentrate on modeling broader contextual relationships. For incorporating global knowledge, KMM introduces a parameterized key–value memory structure that explicitly stores knowledge in addressable slots. This setup decouples knowledge storage from computational pathways, enabling tokens to directly query and retrieve relevant knowledge, thereby enhancing the transparency and flexibility of knowledge access in pretraining. By integrating LFA and KMM, LoKiFormer achieves a more effective fusion of information at both levels, leading to improved representational capacity and adaptability in tackling complex NLP tasks. Our novelty and main contributions are as follows:

- **Local Inductive Bias for Efficient Attention.** We identify that self-attention is inefficient for modeling short-range dependencies. To address this, our LFA introduces a convolutional fusion layer to explicitly capture local patterns before attention, reducing redundant local interactions and allowing attention to focus on more complex and global contextual relationships.

- **Decoupled Parametric Memory for Global Knowledge.** We argue that implicit knowledge representation in MoE couples knowledge storage with computation, limiting transparent access and flexible manipulation. To address this, our KMM explicitly stores reusable knowledge abstractions in addressable slots, enabling direct knowledge retrieval and interpretable inference.

- **Improved Pretraining Efficiency**. Our architecture achieves $1.33\times$ faster convergence during pretraining, reaching the same validation loss in 7.5K steps compared to 10K steps for the baseline. This shows superior optimization efficiency, leading to reduced computational cost and energy consumption.

---

[1]LoKiFormer's internal code name is AIGCoder.

## 2. Related Work

**Efficient Attention Mechanisms**. Recent advances aim to mitigate the quadratic complexity of standard self-attention by approximating full attention through sparse token interactions. Static approaches (Child et al., 2019) employ predefined and input-agnostic patterns, such as local windows (Beltagy et al., 2020), random connections (Zaheer et al., 2020), and global tokens (Xiao et al., 2024). Dynamic sparse attention (Jiang et al., 2024b) adapts the attention pattern to the input content, enabling context-aware computation through token-level pruning (Ren et al., 2023), heavy-hitter selection (Zhang et al., 2023), dynamic group aggregation (Zhang et al., 2025), or global-local fusion (Chen et al., 2025). Despite their effectiveness in lowering FLOPs or memory usage, these methods primarily aim at global computation reduction. In contrast, our proposed LFA does not reduce sequence-level complexity but introduces an explicit local inductive bias, enabling more efficient modeling of short-range dependencies without altering the full attention.

**Memory-Augmented LLMs**. To overcome the limitations of fixed context windows, memory-augmented approaches integrate external knowledge through distinct paradigms. Explicit memory relies on symbolic retrieval from human-readable stores, utilizing text summaries (Zhong et al., 2024), structured knowledge graphs (Jimenez Gutierrez et al., 2024; Modarressi et al., 2025; Chhikara et al., 2025), or database queries (Hu et al., 2023; Packer et al., 2023) to manage vast information. In contrast, implicit memory encodes historical context into compact latent vectors, extending attention capabilities via approximate kNN search (Wu et al., 2022; Wang et al., 2023) or dynamic hierarchical storage pools (Wang et al., 2024; 2025b). While these methods primarily target context extension or online dynamic updates, our KMM focuses on the architectural decoupling of knowledge storage from computation. Rather than relying on retrieval or online editing, KMM embeds domain knowledge into learnable key–value fields that are trained end-to-end and remain static during inference.

**Mixture-of-Experts**. MoE enables scalable language modeling by activating only a subset of experts per token, thus increasing model capacity without proportional computational cost (Shazeer et al., 2017; Fedus et al., 2022). Recent work improves efficiency through better routing strategies, such as load-balanced gating (Fedus et al., 2022), expert choice mechanisms (Zhou et al., 2022), and stable training formulations (Dai et al., 2022; Pan et al., 2024). Model initialization has also been simplified by constructing experts from dense checkpoints (Komatsuzaki et al., 2023; Wei et al., 2024). Despite these advances, knowledge in MoE remains implicitly encoded within parameters. In contrast, our method introduces an explicit, parameterized key-value store that supports direct querying and reusable knowledge.

## 3. Background: Multi-Head Latent Attention

Recently, DeepSeek (Liu et al., 2024b) introduces multi-head latent attention (MLA) mechanism, which compresses contextual representations into a lower-dimensional latent space to improve the efficiency of long-context modeling. Given an input sequence $\mathbf{U}=\{\mathbf{u}_1, \mathbf{u}_2, \ldots, \mathbf{u}_L\} \in \mathbb{R}^{L \times d}$ of $L$ tokens with dimension $d$, MLA first projects each token into a compact latent representation:

$$
\begin{aligned}
\mathbf{c}^{\mathrm{Q}} &= \mathbf{U}\mathbf{W}^{\mathrm{QD}}, \quad \mathbf{c}^{\mathrm{Q}} \in \mathbb{R}^{L \times d'_c}, \quad d'_c < d, \\
\mathbf{c}^{\mathrm{KV}} &= \mathbf{U}\mathbf{W}^{\mathrm{KVD}}, \quad \mathbf{c}^{\mathrm{KV}} \in \mathbb{R}^{L \times d_c}, \quad d_c < d,
\end{aligned}
\tag{1}
$$

where $\mathbf{W}^{\mathrm{QD}} \in \mathbb{R}^{d \times d'_c}$ and $\mathbf{W}^{\mathrm{KVD}} \in \mathbb{R}^{d \times d_c}$ are the down-projection matrices. The latent representation is then expanded to obtain query, key, and value through $h$ parallel linear projection. Finally, the multi-head latent attention is computed as via $h$ parallel attention heads:

$$
\begin{aligned}
\mathrm{MLA}(\mathbf{U}) &= [\mathbf{Att}^1, \ldots, \mathbf{Att}^h] \in \mathbb{R}^{L \times d}, \\
\mathbf{Att}^i &= \mathrm{softmax}\left(\frac{\mathbf{Q}^i \mathbf{K}^{i\top}}{\sqrt{d_h}}\right)\mathbf{V}^i, \\
\mathbf{Q}^i &= \mathbf{c}^{\mathrm{Q}}\mathbf{W}^{Q_i}, \; \mathbf{K}^i = \mathbf{c}^{\mathrm{KV}}\mathbf{W}^{K_i}, \; \mathbf{V}^i = \mathbf{c}^{\mathrm{KV}}\mathbf{W}^{V_i},
\end{aligned}
\tag{2}
$$

with $\mathbf{W}^{Q_i} \in \mathbb{R}^{d'_c \times d_h}, \mathbf{W}^{K_i} \in \mathbb{R}^{d_c \times d_h}, \mathbf{W}^{V_i} \in \mathbb{R}^{d_c \times d_h}$ as learnable parameters, $d_h$ denotes the token dimension in each head, $i$ denotes the index of heads. For clarity, we omit the details of rotary position embeddings (RoPE) (Su et al., 2024) in the above formulation. In practice, our implementation follows the same RoPE integration as in the original DeepSeek work (Liu et al., 2024b), and this simplification is made solely for ease of presentation. It is worth noting that, similar to vanilla self-attention, MLA still models short- and long-range dependencies in a undifferentiated manner, which may limit its efficiency in capturing fine-grained local patterns.

## 4. Proposed Method

### 4.1. Architecture Overview of LoKiFormer

In this paper, we propose **LoKiFormer**, a novel LLM architecture for efficient pretraining, which enhances the vanilla decoder block with two complementary modules: Local Fusion Attention (LFA) and Knowledge Memory Module (KMM). These together improve the effectiveness and efficiency of information exploitation at both local and global levels. Our LFA extends the Multi-Head Latent Attention (MLA) (Liu et al., 2024b) framework by introducing an explicit local inductive bias (c.f. Section 4.2). Instead of relying on pairwise interactions, LFA applies convolutional fusion to adjacent tokens before attention. This enables more efficient capture of short-range dependencies while providing the attention with richer contextual representations for modeling broader contextual relationships.

---

**Algorithm 1** Computation Flow of our LoKiFormer Block.

**Require:** Inputs $\mathbf{U} = \{\mathbf{u}_1, \ldots, \mathbf{u}_L\} \in \mathbb{R}^{L \times d}$; knowledge field matrices $\mathcal{K}$ and $\mathcal{V}$; learnable parameters $\mathbf{W}^{\mathrm{QD}}, \mathbf{W}^{\mathrm{KVD}}, \mathbf{W}^{\mathrm{Q}}, \mathbf{W}^{\mathrm{K}}, \mathbf{W}^{\mathrm{V}}, \mathbf{W}^{\mathrm{H}}, \mathbf{W}^{\mathrm{O}}$; hyperparameters $h$, $k$ and $c$.
1: Compute locally fused representation $\hat{\mathbf{U}} = \mathrm{Conv}(\mathbf{U})$ via Eqn. (3) with kernel size $k$ and group count $h$.
2: Compute $\mathbf{c}^{\mathrm{Q}} = \hat{\mathbf{U}}\mathbf{W}^{\mathrm{QD}} \in \mathbb{R}^{L \times d'_c}, \mathbf{c}^{\mathrm{KV}} = \hat{\mathbf{U}}\mathbf{W}^{\mathrm{KVD}} \in \mathbb{R}^{L \times d_c}$.
3: Compute $\mathbf{O}^{\mathrm{A}} = \mathrm{MLA}(\mathbf{c}^{\mathrm{Q}}, \mathbf{c}^{\mathrm{KV}}; \mathbf{W}^{\mathrm{Q}}, \mathbf{W}^{\mathrm{K}}, \mathbf{W}^{\mathrm{V}})$ with $h$ heads.
4: Compute $\mathbf{H} = \mathbf{c}^{\mathrm{KV}}\mathbf{W}^{\mathrm{H}}, \mathbf{Z} = \mathrm{KMM}(\mathbf{H}; \mathcal{K}, \mathcal{V}), \mathbf{O}^{\mathrm{K}} = \mathbf{Z}\mathbf{W}^{\mathrm{O}}$.
5: Compute expert-enhanced output: $\mathbf{O} = \mathrm{MoE}(\mathbf{O}^{\mathrm{A}} + \mathbf{O}^{\mathrm{K}})$.
**Ensure:** Final block output $\mathbf{O}$.

---

At the global level, our KMM provides an explicit key–value memory for knowledge access. It takes the latent representation $\mathbf{c}$ as input, projects it into queries, matches them against learnable keys that encode commonsense and domain-specific knowledge fields. Subsequently, we aggregate the corresponding values to produce the output (c.f. Section 4.3). Finally, the output of KMM is combined with the MoE result to form the final decoder-block representation. In this case, KMM augments the implicit parameterization of MoEs with an explicit, reusable memory pathway, improving transparency and flexibility in knowledge access. The overview and pseudo code of our proposed architecture are shown in Figure 1 and Algorithm 1, respctively.

### 4.2. Locality-aware Attention via Convolutional Fusion

Natural language exhibits strong locality, where adjacent tokens are highly correlated at the semantic levels. Prior works (Chen et al., 2025) have demonstrated that local modeling improves representation performance. However, both vanilla self-attention (Vaswani et al., 2017) and multi-head latent attention (MLA) (Liu et al., 2024b) model short-range dependencies together with long-range ones, requiring global pairwise interactions even when only local patterns are needed. This may be inefficient during pretraining for capturing local fine-grained relationships between adjacent tokens. To alleviate this issue, we propose **Local Fusion Attention (LFA)**, which enhances the attention by introducing a group convolution that explicitly fuses features of neighboring tokens (validated in ablation Figure 3). This provides a strong local inductive bias, enabling more efficient modeling of short-range dependencies while preparing better contextualized representations.

Formally, we split the hidden states $\mathbf{U} \in \mathbb{R}^{L \times d}$ along the feature dimension into $h$ groups, each of size $d_h = d/h$: $\mathbf{U} = [\mathbf{U}^{(1)}, \mathbf{U}^{(2)}, \cdots, \mathbf{U}^{(h)}]$, where $\mathbf{U}^{(g)} \in \mathbb{R}^{L \times d_h}$ denotes the hidden states for the group $g$. The number of groups is set equal to the number of attention heads $h$, so that each head receives its own locally fused representation and learn distinct local fusion patterns. For each group $g \in [1, h]$, we apply a causal 1D convolution with a kernel

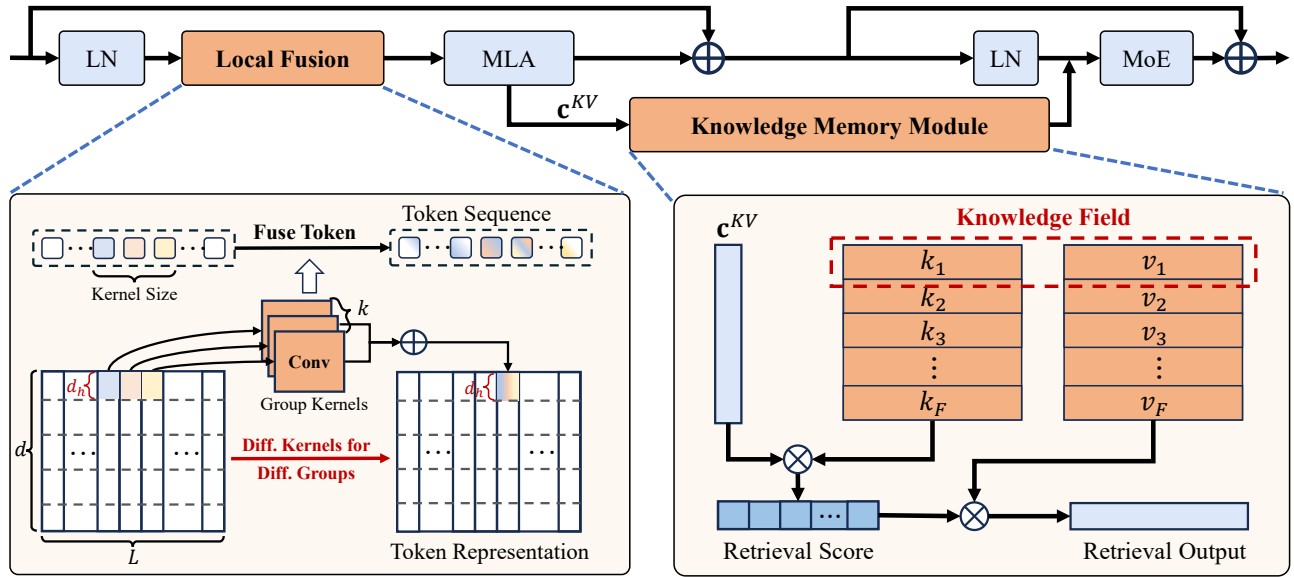

*Figure 1.* Overall architecture of the decoder layer in our LoKiFormer. The model enhances a vanilla decoder block with two proposed modules: Local Fusion Attention (LFA) and the Knowledge Memory Module (KMM). Together with the Multi-Head Latent Attention (MLA) backbone and Mixture-of-Experts (MoE) layer, these components provide complementary local and global modeling capabilities, improving both modeling efficiency and knowledge exploitation.

$\Theta^{(g)} \in \mathbb{R}^{k \times d_h \times d_h}$ along the sequence dimension to aggregate information from its $k$ immediate neighbors within the same group, creating a richer, context-aware representation before being projected into the latent space for attention. At position $t$, the aggregation involves only tokens in the range $[t - k + 1, t]$ (with left padding if needed). This ensures no access to future tokens, maintaining causal masking in accordance with the standard attention mechanism. The fused representation for group $g$ is computed as:

$$\hat{\mathbf{U}}_t^{(g)} = \sum_{s=0}^{k-1} \mathbf{U}_{t-s}^{(g)} \cdot \Theta^{(g)}[s], \qquad (3)$$

where $\Theta^{(g)}[s] \in \mathbb{R}^{d_h \times d_h}$ is the slice of the kernel at position $s$. The index $t - s$ explicitly moves along the sequence axis ($L$), meaning that each output position aggregates information from its previous $k$ neighboring tokens within the same group. Finally, we concatenate all groups to obtain $\hat{\mathbf{U}} = [\hat{\mathbf{U}}^{(1)}, \hat{\mathbf{U}}^{(2)}, \cdots, \hat{\mathbf{U}}^{(h)}]$. Note that we use a stride of 1 and apply suitable padding so that the sequence length remains unchanged, *i.e.*, $\hat{\mathbf{U}}^{(g)} \in \mathbb{R}^{L \times d_h}$. The locally fused representation $\tilde{\mathbf{U}}$ is then used for query, key and value construction via Eqn.(1). Our proposed LFA is a lightweight modification only introduces a small number of additional parameters (only 0.41% in our LoKiFormer-7B model). Empirical results show that LFA accelerates convergence in training (c.f. Figure 3) and consistently enhances downstream performance (c.f. Tables 2 and 3), demonstrating its effectiveness despite the minimal overhead.

### 4.3. Explicit Parametric Memory for Global Knowledge

While the proposed LFA focuses on enhancing the modeling of short-range dependencies, effectively leveraging global-level knowledge remains a challenge. Existing approaches such as MoE layers (Shazeer et al., 2017) implicitly encode knowledge within distributed parameters, which must be re-activated through linear transformation for every query. This implicit storage lacks transparency and restricts efficient reuse of knowledge patterns. To address this and improve knowledge internalization during pretraining, we propose a **Knowledge Memory Module (KMM)**, which explicitly organizes global knowledge into parameterized key–value fields that can be directly queried during model execution (validated in ablation Figure 3).

Instead of directly using the original hidden states $\mathbf{U}$, KMM takes the latent representation $\mathbf{c}^{\text{KV}} \in \mathbb{R}^{L \times d_c}$ from MLA as input. This compact representation is both computationally efficient and semantically rich, making it well-suited for querying global knowledge fields. We first project it to query space with learnable parameters $\mathbf{W}^{\text{H}} \in \mathbb{R}^{d_c \times d_u}$:

$$\mathbf{H} = \mathbf{c}^{\text{KV}} \mathbf{W}^{\text{H}}, \qquad (4)$$

where $\mathbf{H} \in \mathbb{R}^{L \times d_u}$ denotes the query features projected from the latent KV representation $\mathbf{c}^{KV}$.

**Multi-group knowledge retrieval over parameterized fields.** Our KMM adopts a multi-group retrieval scheme to increase capacity. Formally, we store the knowledge fields

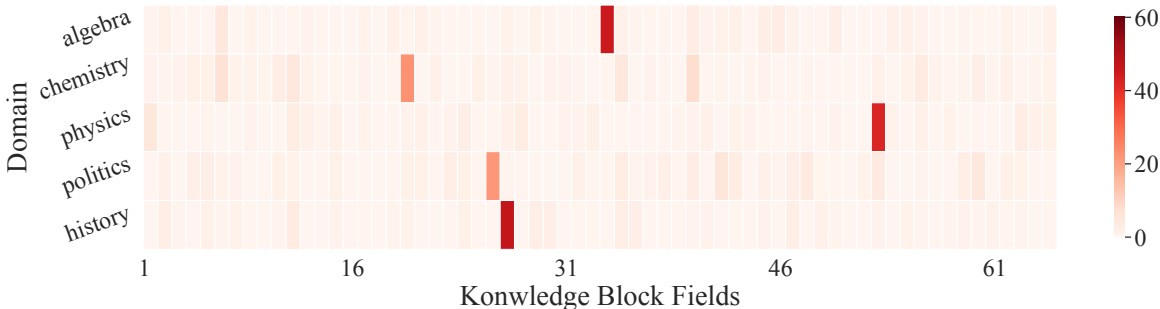

*Figure 2.* Field–domain associations of the proposed KMM with 64 knowledge fields, evaluated on five MMLU domains (1,024 samples each). The heatmap of softmax($\mathbf{H}\mathcal{K}/d_u$) in layer 4 shows that fields emerge with domain-specific specialization, enabling explicit and interpretable knowledge retrieval. We put more visualizations of the remaining layers in Section D of the supplementary.

as parameterized keys and values:

$$\mathcal{K} \in \mathbb{R}^{F \times d_u}, \quad \mathcal{V} \in \mathbb{R}^{F \times d_v}, \tag{5}$$

where $F$ denotes the number of fields, $d_u$ is the query/key dimension, and $d_v$ is the value dimension. Note that $\mathcal{K}$ and $\mathcal{V}$ are randomly initialized and optimized end-to-end during training. We split the projected query $\mathbf{H}$ and the knowledge field parameters $\mathcal{K}, \mathcal{V}$ along the feature dimension into $c$ groups, where each group corresponds to a lower-dimensional subspace. In each group, KMM computes the similarity between the group-specific query $\mathbf{H}^{(i)}$ and the knowledge keys $\mathcal{K}^{(i)}$ using scaled dot-product, and then aggregates the corresponding values $\mathcal{V}^{(i)}$ to obtain the retrieved representation $\mathbf{Z}^{(i)}$. The outputs from all groups are finally concatenated to form the overall representation:

$$\text{KMM}(\mathbf{H}; \mathcal{K}, \mathcal{V}) = [\mathbf{Z}^{(1)}, \dots, \mathbf{Z}^{(c)}] \in \mathbb{R}^{L \times d},$$
$$\mathbf{Z}^{(i)} = \text{softmax}\left(\frac{\mathbf{H}^{(i)} \mathcal{K}^{(i)\top}}{\sqrt{d_u}}\right) \mathcal{V}^{(i)}. \tag{6}$$

The knowledge-enhanced output in Eqn. (6) then will be projected back to the model dimension through a linear transformation with learnable parameters $\mathbf{W}^{\mathrm{O}} \in \mathbb{R}^{d_v \times d}$, yielding $\mathbf{O}^{\mathrm{K}}$. This representation is then combined with the output of the MLA, denoted as $\mathbf{O}^{\mathrm{A}}$, by simple addition: $\mathbf{O}^{\mathrm{A}} + \mathbf{O}^{\mathrm{K}}$. The combined representation is subsequently processed by MoE layer[2], producing the final output: $\mathbf{O} = \text{MOE}(\mathbf{O}^{\mathrm{A}} + \mathbf{O}^{\mathrm{K}})$. In this way, the final output integrates the generative expressiveness of MoE with the explicit knowledge reuse enabled by KMM, allowing the model to balance the creation of novel patterns with the efficient retrieval of knowledge fields.

During inference, the knowledge fields are fixed but their usage is still query-dependent. Different inputs retrieve different knowledge fields based on Eqn.(6), allowing the model to adapt to a wide range of queries without modifying

the memory itself. This decoupling of knowledge from computation enhances the interpretability and modularity.

**Differences of KMM from Attention.** The roles of KMM and attention are fundamentally different. In attention mechanisms, both keys and values are dynamically computed from the input at each step, allowing the model to capture contextual dependencies within the sequence. This process is flexible but computationally expensive, as it requires full pairwise interactions for each token, making it inefficient for longer sequences. In contrast, KMM uses fixed, parameterized keys and values for knowledge retrieval, which are stored in dedicated memory fields. This allows KMM to provide explicit access to pre-learned knowledge abstractions that are decoupled from the input sequence, enabling efficient knowledge reuse. While attention models dependencies within a sequence, KMM enables the reuse of global knowledge across tasks without recalculating context each time. By decoupling knowledge storage from computation, KMM enhances both the flexibility and interpretability.

**Visualization of Knowledge Field-Domain Associations**. To provide an initial validation of our design, we conduct a qualitative analysis by examining the model's behavior on five diverse domains in MMLU (Hendrycks et al., 2021) benchmark. In Figure 2, the results demonstrates that our KMM facilitates the emergence of semantically specialized knowledge fields. The analysis reveals that, through end-to-end training, individual fields spontaneously evolve to represent concepts from specific domains (*e.g.*, Field 25 is consistently activated for history-related questions, while Field 51 is specialized for physics). The specialization is statistically robust, as evidenced by consistent activation patterns aggregated across multiple layers and a large number of samples. This observed self-organization into a structured memory is a key advantage of our explicit design over implicit approaches like MoE. This enables targeted knowledge retrieval, much like consulting an expert. This makes the model's predictions more interpretable and reliable, as the knowledge source is directly observable.

---

[2]Details of the employed MoE architecture are provided in the supplementary material.

# 5. Experiments

## 5.1. Experimental Setup

**Models**. Based on the LoKiFormer architecture, we build a family of models with parameter sizes 1B, 5B, 7B, 13B, 33B, and 60B. This enables a systematic evaluation of the effectiveness of LoKiFormer across model capacities. Unless otherwise specified, our main experiments are conducted on LoKiFormer-7B. Detailed architectural configurations are provided in Section B of the supplementary.

**Metric**. We evaluate our model across four core dimensions using standard benchmarks. For language understanding, we report results on MMLU (Hendrycks et al., 2021) for English, and CMMLU (Li et al., 2024), C-Eval (Huang et al., 2023) for Chinese. Reasoning is assessed with HellaSwag (Zellers et al., 2019) and ARC-Challenge (Clark et al., 2018). We measure code generation performance via pass@1 on HumanEval (Chen et al., 2021), and evaluate mathematical reasoning using GSM8K (Cobbe et al., 2021). We put more details in Section C.1 in the supplementary.

**Baseline Models**. We compare our LoKiFormer-7B with: i) size-comparable models, including Llama-3.1-8B (Grattafiori et al., 2024), Qwen2.5-7B (Yang et al., 2024), Gemma-7B (Team et al., 2024b), InternLM2-7B (Cai et al., 2024), Phi-3-medium (Abdin et al., 2024), Mixtral-8×7B (Jiang et al., 2024a), and DeepSeek-MoE-16B (Dai et al., 2024); ii) frontier open-source models, including Llama-3.1-70B, Llama-3.1-405B (Grattafiori et al., 2024), Qwen2.5-72B (Yang et al., 2024), Mixtral-8×22B (Jiang et al., 2024a), DeepSeek-V2.5 (Liu et al., 2024a), and Hunyuan-Large (Sun et al., 2024); and iii) frontier closed-source models, including Claude-3.5-Sonnet-1022 (Anthropic, 2024) and Gemini-1.5-Pro (Team et al., 2024a). All evaluations are performed with **instructed versions**.

**Implementation Details**.[3] We pretrain our model on the publicly released Matrix Data Pile, a comprehensive bilingual corpus of 4.5T high-quality tokens curated for the MAP-Neo series (Zhang et al., 2024), comprising re-processed high-quality English datasets (*e.g.*, RedPajama (Weber et al., 2024), Dolma (Soldaini et al., 2024)) and Chinese datasets (*e.g.*, Skypile (Wei et al., 2023), ChineseWebText (Chen et al., 2023)), along with a large volume of newly crawled Chinese web content. For validation, we randomly sample 1% from the English Common Crawl portions of the Matrix Data Pile to form validation sets of approximately 18B. For SFT, we employ a curated dataset of approximately 10B high-quality tokens, collected and filtered from existing public instruction-following corpora. We put the details of the pretraining and SFT dataset in Sections C.1 and C.2, respectively.

---

[3]The source code for this project is publicly available at https://github.com/zliu69/aigcode_lokiformer.

We pretrain the models with the Megatron-LM framework on clusters equipped with NVIDIA H200 or Ascend 910B GPUs. For the ablations in Section 5.4, we use 5B model to investigate the effectiveness of key components. In this setting, we employ a global batch size of 1,024 and a context length of 2048. We train the model for 10k steps, consuming approximately 21B tokens. For the main pre-training runs, the 7B and 13B models use a global batch size of 16,384 and a 2048 context for 134k steps (about 4.5T tokens), while the 33B and 60B models employ a 4096 context with the same total tokens. We put more details of the hyperparameters in Section C.3. The 7B model in our evaluations undergoes a two-stage training process. After pre-training, it is fine-tuned via SFT without reinforcement learning.

## 5.2. Demonstration of Performance Gain in Pretraining

**Improved Pretraining Convergence with Proposed Components**. We investigate the effect of different components by progressively add our proposed modules in the baseline. In Figure 3, at 10k training steps, the PPL of the baseline, baseline+LFA, and baseline+LFA+KMM (*i.e.*, our LoKiFormer) are 31.82, 30.57, and 28.50 on the training set, and 31.82, 30.88, and 29.08 on the evaluation set, respectively. Using the baseline's 10k-step PPL (31.82) as a reference, the LFA-augmented model reaches this level at 9k steps, showing 1.11× faster convergence. With both LFA and KMM, our LoKiFormer achieves the same target at 7.8k (train) and 7.5k (eval) steps, corresponding to 1.28× and 1.33× faster convergence. We also conduct zero-shot downstream ablations on the base model to isolate each module's effect. On MMLU, both LFA and KMM yield clear gains over the baseline: 17.9 → 21.2 (+LFA) and 22.9 (+KMM). Combining both further improves performance (25.7), indicating complementary effects. These results demonstrate that our proposed LFA and KMM improves convergence efficiency.

**Scalability of LoKiFormer across Model Sizes**. We evaluated the scalability of LoKiFormer across model sizes from 1B to 60B parameters. As shown in Figure 4, PPL decreases consistently with larger capacity, from 10.05 at 1B to 5.82 at 60B, confirming smooth scaling behavior. In addition, the results indicates that the proposed components, LFA and KMM, remain effective across scales. Pretraining further exhibited stable optimization dynamics, with maximum gradient norms remaining below 1.0 across all models and no collapse or divergence observed. In Table 1, we further compare performance across the base models of LoKiFormer-7B, LoKiFormer-33B, and LoKiFormer-60B. Larger models show significant improvements: LoKiFormer-33B achieves 4.4 points higher on MMLU and 17.9 points on MMLU-Pro compared to LoKiFormer-7B, while LoKiFormer-60B achieves 84.3 on MMLU and 54.6 on MMLU-Pro. The results above demonstrate that LoKiFormer can be scaled reliably to tens of billions of parameters.

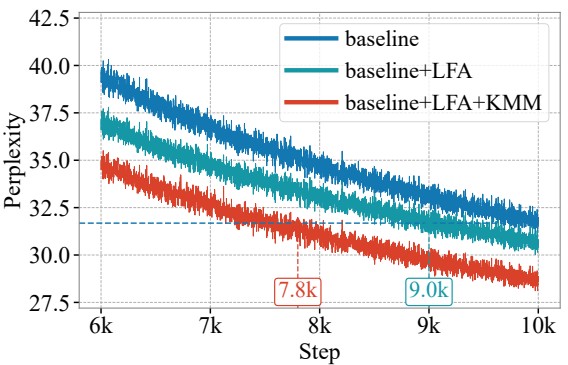 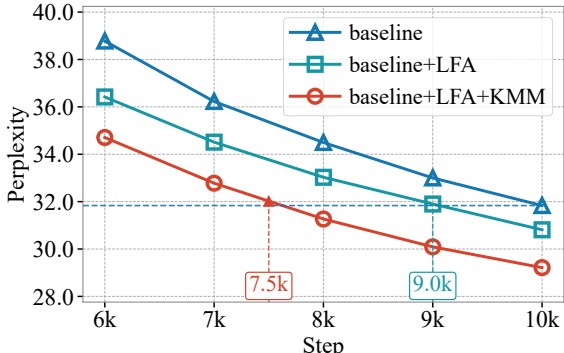

*Figure 3.* Effect of the proposed components LFA and KMM on pretraining performance. Left and right show the training perplexity curves and the evaluation perplexity on the validation set, respectively. All variants are pretrained for 10k steps under the same settings.

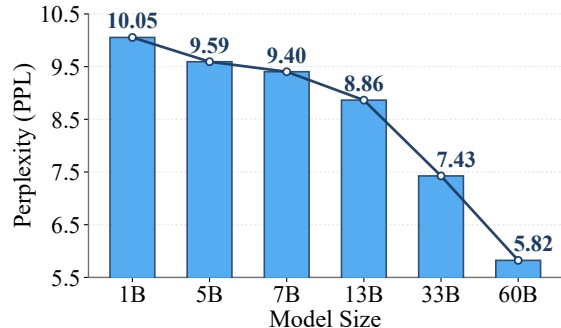

*Figure 4.* Validation perplexity on WikiText-103 across our LoKi-Former models from 1B to 60B parameters.

### 5.3. Performance Comparisons

**Comparisons with Size-comparable Models**. We compare LoKiFormer-7B with size-comparable open-source models across four domains in Table 2. Despite a similar parameter scale, LoKiFormer consistently achieves substantial gains. On language benchmarks, it scores 91.5 on MMLU, 93.4 on CMMLU, and 92.8 on C-Eval, outperforming all dense and MoE baselines by large margins. On reasoning tasks, it attains 89.5 on HellaSwag and 96.2 on ARC-C, again surpassing the strongest size-comparable models. These improvements are mainly attributed to LFA and KMM, which enhance local dependency modeling and explicit knowledge utilization, leading to richer representations during pretraining. Overall, LoKiFormer achieves state-of-the-art performance among size-comparable LLMs.

**Comparisons with Frontier Open-source Models**. We compare LoKiFormer-7B with leading open-source LLMs at much larger scales in Table 3. Despite having only 7B total parameters with 3B activated, our LoKiFormer consistently outperforms these frontier models across most domains. In language understanding, it achieves 91.5 on MMLU, 93.4 on CMMLU, and 92.8 on C-Eval, substantially surpassing the best dense and MoE baselines. Overall,

*Table 1.* Comparisons of various downstream tasks across the base model of our LoKiFormer-7B, 33B and 60B.

| Model | MMLU | MMLU-Pro | CMMLU |
|---|---|---|---|
| LoKiFormer-7B-Base | 74.3 | 18.7 | 72.5 |
| LoKiFormer-33B-Base | 78.7 | 36.6 | 80.0 |
| LoKiFormer-60B-Base | 84.3 | 54.6 | 83.1 |

these results demonstrate the efficiency and effectiveness of our LoKiFormer: with less than one-tenth of the activated parameters, it learns more effectively during pretraining and achieves state-of-the-art performance in most domains, highlighting the benefits of LFA and KMM.

**Comparisons with Frontier Closed-source Models**. Besides, we compare LoKiFormer-7B with frontier closed-source systems, including Claude-3.5-Sonnet and Gemini-1.5-Pro in Table 4. Despite its much smaller scale, LoKiFormer demonstrates highly competitive performance. In language understanding, it achieves 91.5 on MMLU, outperforming both Claude-3.5-Sonnet (88.3) and Gemini-1.5-Pro (85.9) by large margins. These highlight that LoKiFormer can deliver competitive or superior performance to frontier closed-source systems on several challenging benchmarks, despite operating at a fraction of their parameter scale.

### 5.4. Further Experiments

**Knowledge Field Editability Analysis**. To verify that KMM stores explicit and editable domain-level knowledge, we perform knowledge-field removal experiments. For selected fields in Layer 4 (visualized in Figure 2), we zero out their parameters and evaluate performance on five representative MMLU domains. In Figure 5, removing a field mainly degrades its corresponding domain (*e.g.*, algebra drops by 31.8% when Field 32 is removed), indicating that KMM organizes knowledge into domain-specialized slots. Cross-domain effects align with conceptual relatedness: politics

*Table 2.* Comparisons of LoKiFormer-7B with **size-comparable LLMs** across four domains, including language, reasoning, code, math. **Bold** and underlined numbers indicate the best and second-best results, respectively..

| Model | Arch. | # Act. | # Total | Language | | | Reasoning | | Code | Math |
|---|---|---|---|---|---|---|---|---|---|---|
| | | | | MMLU | CMMLU | C-Eval | HellaSwag | ARC-C | HumanEval | GSM8K |
| Llama-3.1-8B | Dense | 8B | 8B | 73.0 | – | 52.0 | 82.3 | 83.4 | 72.6 | 84.5 |
| Qwen2.5-7B | Dense | 7B | 7B | 76.6 | 79.1 | 76.2 | 81.5 | 63.4 | 84.8 | **91.6** |
| Gemma-7B | Dense | 7B | 7B | 64.3 | – | – | 81.2 | 53.2 | 32.3 | 46.4 |
| InternLM2-7B | Dense | 7B | 7B | 63.7 | 63.0 | 60.8 | 83.0 | – | 59.2 | 70.7 |
| Phi-3 medium | Dense | 3.8B | 3.8B | 78.0 | – | – | 82.4 | 91.6 | 62.2 | 91.0 |
| Mixtral-8×7B | MoE | 14B | 46.7B | 70.5 | – | – | 70.4 | 87.3 | 37.8 | 64.7 |
| DeepseekMoE-16B | MoE | 2.4B | 16B | 47.2 | 49.3 | 40.0 | 72.2 | 50.0 | 45.7 | 62.2 |
| LoKiFormer-7B (Ours) | MoE | 3B | 7B | **91.5** | **93.4** | **92.8** | **89.5** | **96.2** | **87.0** | 73.0 |

*Table 3.* Comparisons of LoKiFormer-7B with **frontier open-source LLMs** across four domains, including language, reasoning, code, math. **Bold** and underlined numbers indicate the best and second-best results, respectively.

| Model | Arch. | # Act. | # Total | Language | | | Reasoning | | Code | Math |
|---|---|---|---|---|---|---|---|---|---|---|
| | | | | MMLU | CMMLU | C-Eval | HellaSwag | ARC-C | HumanEval | GSM8K |
| Llama-3.1-70B | Dense | 70B | 70B | 83.6 | 69.0 | – | 86.7 | 94.8 | 80.5 | 95.1 |
| Llama-3.1-405B | Dense | 405B | 405B | 86.0 | – | 61.5 | 88.3 | **96.9** | 89.0 | **96.8** |
| Qwen2.5-72B | Dense | 72B | 72B | 84.4 | 86.7 | 84.7 | – | – | 86.6 | 95.8 |
| Mixtral-8×22B | MoE | 39B | 141B | 77.8 | 61.0 | 60.0 | 89.2 | 90.0 | 75.0 | 85.0 |
| Deepseek-V2.5 | MoE | 21B | 236B | 80.4 | – | – | 85.0 | 72.6 | 89.0 | 88.3 |
| Hunyuan-Large | MoE | 52B | 389B | 89.9 | 90.4 | 89.5 | – | 94.6 | **90.0** | – |
| LoKiFormer-7B (Ours) | MoE | 3B | 7B | **91.5** | **93.4** | **92.8** | **89.5** | 96.2 | 87.0 | 73.0 |

*Table 4.* Comparisons of LoKiFormer-7B with **frontier closed-source LLMs** across four domains. **Bold** and underlined numbers indicate the best and second-best results, respectively. "HellS." and "HumE." are short for "HellaSwag" and "HumanEval".

| Model | MMLU | HellS. | HumE. | GSM8K |
|---|---|---|---|---|
| Claude-3.5-Sonnet-1022 | 88.3 | **94.6** | **92.0** | **96.4** |
| Gemini-1.5-Pro (May'24) | 85.9 | 93.3 | 84.1 | 90.8 |
| LoKiFormer-7B (Ours) | **91.5** | 89.5 | 87.0 | 73.0 |

and history fields affect both domains, while STEM fields exhibit limited mutual influence. These results demonstrate the interpretability and editability of knowledge in KMM.

**Throughput and Latency Analysis**. We compare the training throughput (in tokens per second) of LoKiFormer-7B against the baseline without LFA and KMM under identical hardware configurations (8×NVIDIA A100 GPUs, 80GB memory). At 4K context, the measured throughput for the baseline is 975 tokens/second, while LoKiFormer-7B achieves a throughput of 950 tokens/second. This analysis confirms that the per-step computational overhead introduced by our novel modules is minimal (*i.e.*, 2.6%). Therefore, the significant reduction in the number of training steps required to reach the target loss, translates directly into a substantial reduction in total wall-clock training time. We also

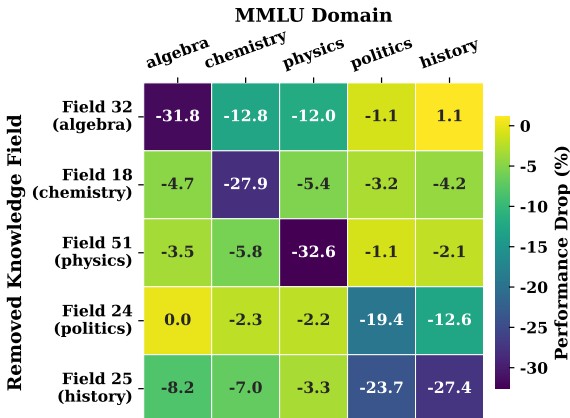

*Figure 5.* Performance drop (%) on five MMLU domains after removing individual knowledge fields in Layer 4. Each field primarily affects its associated domain (diagonal entries), while exhibiting selective cross-domain impacts that reflect semantic relationships between disciplines (*e.g.*, politics/history).

report the inference time of our LoKiFormer-7B on Ascend 910B (8 NPUs). At 4K context, compared with th baseline, decoding throughput remains nearly unchanged (1299 *vs.* 1294 tokens/s/NPU, bs=256), while time-to-first-token increases only slightly (132 → 136 ms). The results above show high computational efficiency of our LoKiFormer.

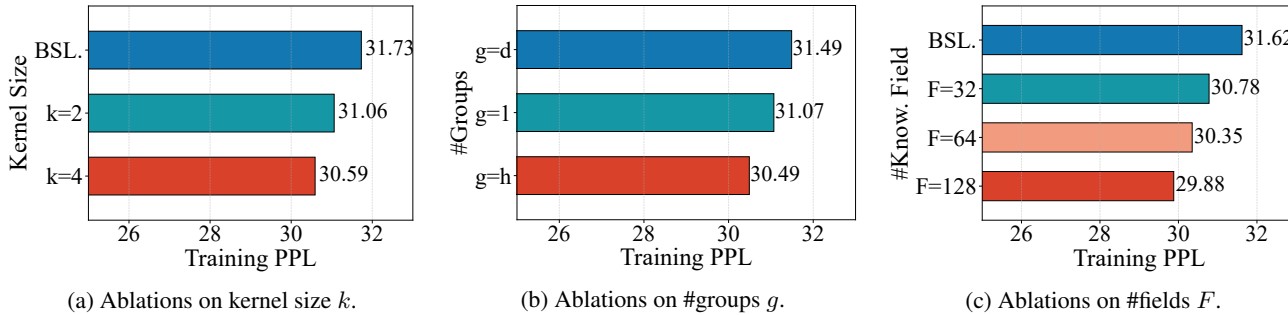

(a) Ablations on kernel size $k$.  (b) Ablations on #groups $g$.  (c) Ablations on #fields $F$.

*Figure 6.* Ablations on the kernel size $k$ and #convolutional group $g$ in LFA, and #knowledge fields $F$. "BSL." refers to "baseline".

**Ablations of Kernel Sizes $k$ in Local Fusion[4].** We ablate the kernel size $k$ of the convolution in the local fusion module with different $k$ (*i.e.*, $k = 2$ and $k = 4$). In Figure 6(a), training PPL decreases from 31.73 (baseline) to 31.06 ($k = 2$) and 30.59 ($k = 4$). Within the tested range, we choose $k = 4$ as a balance between effectiveness and efficiency: it consistently outperforms $k = 2$, while larger values are not explored due to the high cost of large-scale pretraining. This setting yields strong gains with manageable overhead, indicating that an appropriate local receptive field substantially improves pretraining efficiency.

**Ablations of Number of Convolutional Groups $g$ in Local Fusion.** We vary the number of groups $g$ in the convolution of LFA, comparing $g=1$ (no grouping), $g=d$ (grouped by feature dimension), and $g=h$ (aligned with attention heads). Under identical settings, we report training PPL at 10k steps. As shown in Figure 6(b), PPL is 31.07 for $g=1$, 31.49 for $g=d$, and 30.49 for $g=h$. Relative to $g=1$, $g=h$ reduces PPL by 1.87%, while $g=d$ slightly degrades performance; compared to $g=d$, $g=h$ achieves a 3.18% lower PPL. These results indicate that grouping is beneficial in LFA, with head-aligned grouping yielding the best performance.

**Ablations of Number of Knowledge Fields $F$.** We ablate the number of knowledge fields $F$ in KMM. In Figure 6(c), training PPL on the cc-en dataset decreases monotonically as $F$ increases, from 31.62 (without KMM) to 30.78 ($F=32$), 30.35 ($F=64$), and 29.88 ($F=128$). These correspond to relative reductions of 2.66%, 4.02%, and 5.50%, respectively. While $F=128$ achieves the lowest PPL, $F=64$ captures about 73% of the total reduction with lower computational and memory cost. We therefore adopt $F=64$ as a balance between efficiency and performance.

## 6. Conclusion and Future Work

We proposed LoKiFormer, a large language model architecture that augments the standard Transformer decoder with

two complementary modules: LFA for efficient short-range dependency modeling and the KMM for explicit knowledge retrieval. Together, these modules enhance local contextual modeling and global knowledge utilization, resulting in faster pretraining convergence and stronger performance.

In the future, we plan to extend LoKiFormer to multimodal reasoning, such as text, images, and audio. We aim to investigate how LFA can improve local dependency modeling across diverse inputs, while KMM can facilitate the integration and retrieval of knowledge across different modalities. This extension will help us address more complex reasoning tasks, allowing LoKiFormer to leverage both richer contextual and cross-modal knowledge for improved performance.

## Impact Statement

Our work enhance the pretraining efficiency for large language model. By integrating Local Fusion Attention (LFA) and the Knowledge Memory Module (KMM), we enable faster convergence and more effective utilization of both local and global information. This leads to a faster pretraining speed, reducing computational costs and environmental impact. These improvements contribute to the development of more efficient and sustainable AI systems, making large-scale language models more resource-efficient.

## Author Contributions

Qiuwu Chen and Zimo Liu contributed equally to this work. Mingkui Tan and Yaofo Chen are the corresponding authors. Qiuwu Chen and Zimo Liu conceived the main idea and designed the LoKiFormer architecture. Qiuwu Chen, Zimo Liu, Yuchen Li, and Ying Sun developed the method and conducted the experiments. Yifan Zhang, Zhijie Qiu, Zeng You, Ryan Dong, and Simeng Ma contributed to implementation, data preparation, evaluation, and result analysis. Qiuwu Chen and Zimo Liu wrote the draft. Mingkui Tan and Yaofo Chen supervised the project, provided guidance on methodology and experiments, and revised the manuscript.

---

[4]We conduct ablations on LoKiFormer-5B, which is lightweight enough for efficiency while remaining representative.

## Acknowledgments

This work was partially supported by the Joint Funds of the National Natural Science Foundation of China under Grant No.U24A20327, the Guangdong S&T Program under Grant No.2026B0101110001, the GuangDong Basic and Applied Basic Research Foundation under Grant No.2026A1515010388, and the Postdoctoral Fellowship Program of CPSF under Grant No. GZC20251043. We also thank Huawei for their support, providing hardware resources and collaboration opportunities essential to the success of this work.

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

# SUPPLEMENTARY MATERIALS

In the supplementary, we provide more details about our LoKiFormer architecture and more implementation details. We organize our supplementary as follows.

- In Section A, we provide the detailed formulation of our employed MoE architecture.

- In Section B, we provide more details about model specifications and parameter breakdown.

- In Section C, we provide more details of used dataset and implementation.

- In Section D, we provide more empirical analysis to show the effectiveness of our LoKiFormer.

## A. More Details of the Employed MoE Architecture

Below we provide the mathematical formulation of the Mixture-of-Experts (MoE) layer used in our model, which consists of two distinct components: a shared expert that processes all tokens and multiple routed experts that are selectively activated through a gating mechanism, which is similar with DeepseekMoE (Liu et al., 2024a). Given the input hidden states $\mathbf{U} \in \mathbb{R}^{L \times d}$, where $L$ represents the sequence length and $d$ denotes the hidden dimension, the MoE layer processes each token representation $\mathbf{u}_i$ (the $i$-th row of $\mathbf{U}$) independently through the following procedure.

**Formulation for Shared Expert.** For each input token representation $\mathbf{u} \in \mathbb{R}^d$, we first compute the logits for shared expert through:

$$\mathbf{h}^{\mathrm{S}} = \mathrm{MLP}^{\mathrm{S}}(\mathbf{u}), \tag{7}$$

where $\mathrm{MLP}(\cdot)$ denotes a multi-layer perceptron, $\mathbf{h}^{\mathrm{S}}$ remains the same shape, *i.e.*, $\mathbf{h}^{\mathrm{S}} \in \mathbb{R}^d$. Note that in our LoKiFormer, we have only one shared expert. Then a sigmoid-gating mechanism is applied to modulate the shared experts' output, where the gating values are computed as:

$$\mathbf{g}^{\mathrm{S}} = \sigma(\mathbf{W}^{\mathrm{S}}\mathbf{u}), \tag{8}$$

where $\sigma(\cdot)$ denotes the sigmoid function, $\mathbf{g}^{\mathrm{S}}$ has the same shape with $\mathbf{h}^{\mathrm{S}}$, *i.e.*, $\mathbf{g}^{\mathrm{S}} \in \mathbb{R}^d$. The gated output of shared expert is then obtained by:

$$\mathbf{o}^{\mathrm{S}} = \mathbf{g}^{\mathrm{S}} \odot \mathbf{h}^{\mathrm{S}}. \tag{9}$$

**Formulation for Routed Experts.** For each input token representation $\mathbf{u} \in \mathbb{R}^d$, we first compute the router logits through:

$$\mathbf{g}^{\mathrm{R}} = \mathbf{W}^{\mathrm{R}}\mathbf{u}, \tag{10}$$

where $\mathbf{W}^{\mathrm{R}} \in \mathbb{R}^{d \times N_{\mathrm{route}}}$ denotes the router weight matrix, and $\mathbf{g}^{\mathrm{R}} \in \mathbb{R}^{N_{\mathrm{route}}}$ represents the logits for all routed experts. These logits are then passed through a sigmoid function to obtain gating values:

$$\mathbf{g}_{\sigma}^{\mathrm{R}} = \sigma(\mathbf{g}^{\mathrm{R}}), \tag{11}$$

where $\mathbf{g}_{\sigma}^{\mathrm{R}} \in \mathbb{R}^{N_{\mathrm{route}}}$ contains values in the range (0,1). These gating values are used to select the top-k experts for each token:

$$w_i = \frac{g_{\sigma,i}^{\mathrm{R}}}{\sum_{j \in \mathcal{T}} g_{\sigma,j}^{\mathrm{R}}}, \tag{12}$$

where $\mathcal{T}$ denotes the set of top-$k$ selected experts, and $w_i$ represents the normalized routing weight for the $i$-th expert. The output from the routed experts is computed as:

$$\mathbf{o}^{\mathrm{R}} = \left( \sum_{i \in \mathcal{T}} w_i \cdot \mathrm{Expert}_i(\mathbf{u}) \right), \tag{13}$$

where $\mathrm{Expert}_i(\cdot)$ denotes the $i$-th expert network. The final output $\mathbf{o}^{\mathrm{R}} \in \mathbb{R}^d$ has the same dimensionality as the input.

**Integration of Expert Outputs.** The final output for each token is obtained by combining the contributions from both the shared expert and the routed experts. The combined output is computed through element-wise addition:

$$\mathbf{o} = \mathbf{o}^{\mathrm{R}} + \mathbf{o}^{\mathrm{S}}, \tag{14}$$

where $\mathbf{o} \in \mathbb{R}^d$ represents the final output representation for the input token $\mathbf{u}$. This allows the model to benefit from both the consistently applied shared expertise and the specialized processing of the selectively activated routed experts. The complete output hidden states $\mathbf{O} \in \mathbb{R}^{L \times d}$ are constructed by applying this transformation to each token in the sequence independently.

# B. More Details on Model Architecture

**Model Specifications**. In Table 5, we summarize the complete set of hyperparameters that define our model's architecture. We specify key design choices, including the total number of transformer layers, the dimensionality of the hidden states, the number of attention heads in each multi-head attention module, the resultant total parameter count, and the context length used for training.

*Table 5.* Configuration of proposed LoKiFormer models across different scales. $d$, $d_c$ and $k$ denote hidden dimension, MLA latent dimension and LFA kernel size, respectively.

| Model | #Layers | $d$ | $d_c$ | $k$ | KMM ($F \times h \times d_u$) | Context Length | MoE Configuration |
|---|---|---|---|---|---|---|---|
| 1B | 16 | 1024 | 512 | 4 | 64×16×64 | 2048 | 1 shared + 8 experts |
| 5B | 20 | 2048 | 512 | 4 | 64×32×128 | 2048 | 1 shared + 8 experts |
| 7B | 28 | 2048 | 512 | 4 | 64×16×128 | 2048 | 1 shared + 8 experts |
| 13B | 28 | 2048 | 512 | 4 | 64×16×128 | 2048 | 1 shared + 16 experts |
| 33B | 18 | 4096 | 512 | 4 | 64×32×128 | 4096 | 1 shared + 16 experts |
| 60B | 32 | 4096 | 512 | 4 | 64×32×128 | 4096 | 1 shared + 16 experts |

**Component-wise Parameter Breakdown**. In Table 6, we present the distribution of parameters across different LoKiFormer components. The MoE module consistently dominates the parameter budget, especially at larger scales, highlighting its central role in capacity expansion. In contrast, our newly introduced LFA kernel and KMM modules contribute only a marginal fraction of the total parameters, demonstrating the efficiency of these designs in enhancing model expressiveness without incurring substantial overhead. For completeness, we note that the input embedding and LM head are tied in the 1B, 5B, and 7B models, while they are untied in the 13B, 33B, and 60B models, with both components contributing equally when reported separately.

*Table 6.* Distribution of parameters among different LoKiFormer components across different model scales. For the 1B, 5B, and 7B models, the input embedding and the final LM head are tied, hence the LM head column is marked with "–". For the 13B, 33B, and 60B models, the embedding and LM head are untied, and their reported proportions correspond to each individual component.

| Model | Embedding | LFA Kernel | KMM | MLA | MoE | LM Head |
|---|---|---|---|---|---|---|
| 1B | 13.49% | 0.36% | 2.28% | 5.15% | 78.72% | – |
| 5B | 6.03% | 0.41% | 2.09% | 3.49% | 87.98% | – |
| 7B | 4.38% | 0.41% | 2.12% | 3.55% | 89.53% | – |
| 13B | 2.39% | 0.23% | 1.15% | 1.93% | 91.91% | 2.39% |
| 33B | 1.89% | 0.11% | 1.05% | 1.41% | 93.65% | 1.89% |
| 60B | 1.02% | 0.12% | 1.07% | 1.43% | 95.35% | 1.02% |

# C. More Experimental Protocols

## C.1. More Details on Dataset

We pre-train our model on the publicly released Matrix Data Pile, a comprehensive bilingual corpus of 4.5T high-quality tokens curated for the MAP-Neo series (Zhang et al., 2024), comprising re-processed high-quality English datasets (*e.g.*, RedPajama (Weber et al., 2024), Dolma (Soldaini et al., 2024)) and Chinese datasets (*e.g.*, Skypile (Wei et al., 2023), ChineseWebText (Chen et al., 2023)), along with a large volume of newly crawled Chinese web content. For validation, we randomly sample 1% from the English Common Crawl (cc_en) portions of the Matrix Data Pile to form validation sets of approximately 18B. For supervised fine-tuning (SFT), we employ a curated dataset of approximately 10B high-quality tokens, collected and filtered from existing public instruction-following corpora. In the following, we depict the details of the datasets used for pre-training, supervised fine-tuning and evaluation.

**Matrix Data Pile** (Zhang et al., 2024) is a large-scale, 4.5 trillion-token bilingual pre-training corpus meticulously curated for the MAP-Neo model series. The English subset is derived from a re-processing of high-quality public datasets, including RedPajama-Data-V2 (Weber et al., 2024), Dolma (Soldaini et al., 2024), Cultrax (EN) (Nguyen et al., 2024), Amber (Refined-Web) (Liu et al., 2024c), and SlimPajama (Cerebras, 2024), each subjected to a rigorous multi-phase filtering pipeline involving heuristic-based noise removal and deduplication strategies. The Chinese subset is predominantly composed of 80.6% newly crawled web content collected from scratch to address the persistent scarcity of open high-quality

Chinese data; This foundational collection is enriched through integration of existing public Chinese datasets such as ChineseWebText (Chen et al., 2023), Wanjuan (He et al., 2023), Yayi2 (Luo et al., 2023), Cultrax (ZH) (Nguyen et al., 2024), and Skypile (Wei et al., 2023). To further enrich the corpus, high-quality data from diverse domains are incorporated, including programming code, academic papers (e.g., arXiv), books, government reports, and specialized collections for mathematics and science. All constituent data, regardless of origin or language, are processed through a unified framework that applies extensive language-specific cleaning (including HTML artifact removal and OCR error correction tailored for Chinese text) and robust deduplication techniques—encompassing exact document, MinHash LSH, and a proposed similar line deduplication—to ensure consistency, quality, and representativeness across the entire corpus.

**MMLU** (Hendrycks et al., 2021) is a comprehensive benchmark designed to evaluate multitask knowledge and reasoning capabilities of language models across 57 diverse subjects, spanning STEM, social sciences, humanities, and professional domains such as law, medicine, and business. The dataset consists of multiple-choice questions sourced from open-access exam materials, textbooks, and introductory college courses, ensuring broad coverage of both foundational academic principles and specialized domain-specific knowledge. The Questions are carefully curated to require not only factual recall, but also higher-order reasoning, including logical inference and conceptual understanding. To assess zero-shot and few-shot generalization, the benchmark is typically evaluated with minimal in-context examples, making it a robust measure of broad world knowledge and cross-domain transfer ability. Its design emphasizes domain diversity and task heterogeneity, providing a carefully structured assessment of model strengths and limitations across disciplines.

**C-Eval** (Huang et al., 2023) is a comprehensive Chinese benchmark designed to evaluate the knowledge and reasoning capabilities of foundation models across a wide range of academic disciplines. The dataset consists of 13,948 multiple-choice questions spanning 52 subjects, categorized into four broad domains: STEM, humanities, social sciences, and professional fields such as law, medicine, and finance. A key feature of C-Eval is its multi-level difficulty design, with questions sourced from exams at middle school, high school, college, and professional qualification levels, enabling fine-grained assessment of model performance across different expertise tiers. This hierarchical structure allows for a rigorous evaluation of both foundational knowledge and advanced reasoning skills in a Chinese educational context. To ensure data quality and minimize contamination, questions are collected from mock exams and past university tests, processed through careful parsing and manual validation, and formatted consistently with standardized answer choices. The benchmark supports both zero-shot and few-shot evaluation protocols, with a public development set provided for prompting and a private test set scored via an online submission system to maintain integrity. C-Eval also includes a challenging subset, C-Eval Hard, composed of complex subjects like advanced mathematics and physics that demand sophisticated problem-solving abilities. By covering diverse topics and difficulty levels within a culturally relevant framework, C-Eval provides a truly robust and nuanced evaluation of Chinese language understanding and domain-specific knowledge in LLMs.

**CMMLU** (Li et al., 2024) is a comprehensive benchmark designed to evaluate multitask language understanding and knowledge acquisition of large language models in the Chinese linguistic and cultural context. The dataset comprises 11,528 multiple-choice questions that span 67 subjects across four broad categories: STEM, humanities, social sciences, and other domains, including 16 China-specific subjects such as Chinese driving rules, traditional food culture, and civil service exams. This deliberate inclusion of region-specific knowledge ensures that the assessment captures culturally grounded reasoning, distinguishing it from general-purpose multilingual benchmarks. Questions are curated from academic sources and standardized tests, covering difficulty levels from elementary to professional expertise, enabling fine-grained evaluation of model capabilities across diverse knowledge domains. To support the few-shot evaluation, each subject includes a development set of five exemplars, with the remainder forming the test set. CMMLU is specifically structured to assess zero-shot and few-shot generalization, making it a robust measure of a model's ability to transfer knowledge and reason across specialized and culturally relevant topics. Its design emphasizes domain diversity, cultural specificity, and task heterogeneity, providing a rigorous and well-nuanced assessment of Chinese language understanding that complements existing global benchmarks.

**HellaSwag** (Zellers et al., 2019) is designed to assess models' ability to predict future actions within common real-life situations. It comprises 39,905 context-based completion tasks drawn from video descriptions and textual narratives, each presenting four candidate endings—one correct human-authored option and three adversarially created distractors generated by models to closely match the context in plausibility. The challenge of HellaSwag lies in its emphasis on nuanced, grounded commonsense reasoning and its robustness against superficial linguistic cues or heuristic exploitation.

**ARC-Challenge** (Clark et al., 2018) is the more difficult subset of the AI2 Reasoning Challenge (ARC), designed to assess models' ability in nuanced scientific reasoning. It contains 2,590 multiple-choice questions from grade-school science

curricula, filtered to exclude those easily answered by retrieval baselines. The dataset prioritizes tasks demanding deeper cognitive processing, compelling models to go beyond mere fact lookup and instead synthesize prior knowledge and reason about causal or explanatory relationships in scientific phenomena. Its construction emphasizes the distinction between simple information recall and genuine understanding of scientific concepts.

**HumanEval** (Chen et al., 2021) is a widely adopted benchmark designed to evaluate the functional correctness of the generated code through unit test execution. It features 164 carefully hand-crafted programming problems in Python, each accompanied by a function signature, a descriptive docstring, and multiple comprehensive ground-truth test cases that verify the output for various inputs. The evaluation relies on pass@1 metrics, which typically measures the probability that at least one generated solution (out of a single attempt) passes all associated test cases without error. A solution is deemed correct only if it produces semantically accurate results for every test instance, thereby enforcing strict robustness against syntactically valid but logically flawed code, a common failure mode in code generation. By covering algorithmic logic, string manipulation, and mathematical operations, the benchmark assesses a model's ability to generate precise, executable code from natural language specifications, making it a standard for measuring practical coding proficiency in LLMs.

**GSM8K** (Cobbe et al., 2021) is a benchmark for assessing the multi-step mathematical reasoning capabilities of language models through elementary school-level word problems. It contains 7,473 training and 1,319 hand-written test questions, each requiring two to eight sequential reasoning steps. The problems are designed to evaluate not only a model's ability to perform accurate arithmetic computations but also its capacity to decompose complex scenarios into coherent intermediate steps, applying concepts such as proportions, basic algebra, and numerical logic. Characterized by high linguistic diversity and real-world situational contexts, the dataset demands precise comprehension of structured natural language narratives. To facilitate transparent evaluation of reasoning processes, solutions are expected to include explicit chain-of-thought derivations, enabling analysis of both correct logical pathways and systematic errors, rather than predicting final answers.

### C.2. Supervised Fine-Tuning (SFT) Dataset Construction

To ensure high-quality supervision for instruction tuning, we construct a knowledge-intensive SFT dataset derived from large-scale raw corpora. The goal is to distill structured, factual question–answer pairs resembling benchmark tasks such as MMLU (Hendrycks et al., 2021), while rigorously preventing any benchmark leakage. Starting from 6.5 billion raw text segments, we obtain a final corpus of 2.5 million clean and decontaminated samples (average context length is about 4k with padding and masking). The entire pipeline comprises five major stages, as summarized below.

**Stage 1: Rule-based Filtering** (6.5B $\rightarrow$ 50M samples). We first perform large-scale heuristic filtering to remove irrelevant or low-quality content. This step identifies knowledge-oriented and interrogative sentences by matching question patterns (*e.g.*, what, why, which), detecting multiple-choice indicators (A. / B. / (A)(B) *etc.*), and applying keyword-based selection over 57+ academic domains (physics, law, medicine, history, etc.). Creative, subjective, or conversational text is excluded. Approximately 50 million candidate Q&A fragments remain after this stage.

**Stage 2: Semantic Retrieval Filtering** (50M $\rightarrow$ 5M samples). To retain samples that are related to benchmark-style knowledge, we apply an embedding-based retrieval filter. i) We first construct a knowledge seed query set by extracting 10,000 exam-style questions from MMLU, CMMLU, CEval, and GAOKAO, then distilling only their underlying knowledge concepts (*e.g.*, formula of Newton's second law) instead of reusing question text. ii) Both seed queries and candidate samples are encoded using the *bge-large-zh-v1.5* model, and top-k samples with cosine similarity $> 0.65$ are retained. **This retrieval step ensures topical relevance without reusing any benchmark text**.

**Stage 3: LLM-based Structuring and Validation** (5M $\rightarrow$ 3M samples). In this stage, an LLM (*e.g.*, Qwen-Max) is prompted to validate and normalize each candidate fragment into a structured format containing {question, answer, subject, type}. The model verifies factual correctness, conciseness, and clarity, discarding vague or subjective responses. Optional distractor options are added for multiple-choice questions. This yields approximately 3 million high-quality QA samples.

**Stage 4: Balancing and Standardization** (3M $\rightarrow$ 2.8M samples). We perform stratified sampling to balance the subject distribution according to the proportions in MMLU and CMMLU. Underrepresented disciplines such as ethics and nutrition are slightly oversampled. All samples are standardized into a consistent instruction–input–output format and de-duplicated using MinHash and LSH. Personally identifiable information (PII) is also filtered. The resulting dataset contains about 2.8 million balanced and clean samples.

**Stage 5: Benchmark Decontamination** (2.8M $\rightarrow$ 2.5M samples). We employ a rigorous three-fold process to eliminate any potential benchmark leakage.

*Table 7.* Comparisons of zero-shot performance of pretrained base models (without SFT) in various downstream tasks.

| Model | MMLU | CMMLU | C-Eval | HellaSwag | ARC-C |
|---|---|---|---|---|---|
| DeepSeek-7B | 48.2 | 47.2 | 45.0 | 75.4 | – |
| MAP-Neo-7B | 58.1 | 55.1 | 57.7 | 70.7 | 68.1 |
| Qwen2.5-7B | 74.2 | – | – | 80.2 | 63.7 |
| Gemma-7B | 64.3 | – | – | 81.2 | 53.2 |
| InternLM2-7B | 65.8 | 66.3 | 65.8 | 79.3 | – |
| Mixtral-8×7B | 70.6 | 53.3 | 55.4 | 84.4 | 59.7 |
| LoKiFormer-7B (Ours) | **74.3** | **72.5** | **65.8** | **90.8** | **79.4** |

*Table 8.* Compassions between MAP-Neo-7B and our LoKiFormer-7B trained with the same SFT dataset.

| Model | MMLU | CMMLU |
|---|---|---|
| MAP-Neo-7B | 64.4 | 68.6 |
| LoKiFormer-7B | **91.5** | **93.4** |

- **Literal Matching**: remove exact or near-duplicate overlaps (edit distance $\leq 5$ or Jaccard similarity $\geq 0.95$) with a blacklist containing all samples from MMLU, CMMLU, CEval, GAOKAO, and AGIEval (about 20–30k items).

- **Semantic Matching**: compute cosine similarity between encoded blacklist items and candidates; remove those with similarity $\geq 0.92$.

- **Knowledge Triplet Matching**: extract structured knowledge triples (entity, relation, value) using an LLM and flag samples that share identical triples with blacklist items as high-risk candidates for benchmark leakage. These high-risk samples are then either manually audited or conservatively removed. As an auxiliary probe, we also perform generative leakage detection with a smaller trained model, which flags samples whose predicted answers match the ground-truth with extremely high confidence (e.g., probability $> 0.99$) as additional high-risk candidates. Finally, a small-scale manual audit of 1,000 random samples confirms a residual leakage rate below 0.1%. The resulting dataset contains 2.5 million high-quality SFT samples.

This multi-stage pipeline has been validated across several large-model projects and demonstrates strong practicality: it efficiently distills high-quality, knowledge-focused QA data while meeting the strict decontamination and transparency standards required for academic evaluation and open-source release. It provides a clean and reliable basis for the supervised fine-tuning phase of our LoKiFormer.

**More Discussions of the SFT Dataset.** In Stage 2, the semantic retrieval filtering may inadvertently lead to data being overly specialized towards specific knowledge domains, potentially biasing the dataset. To address this concern, we conducted two additional experiments. First, we evaluate and compare the zero-shot performance of several base pretrained models without fine-tuning. As shown in Table 7, LoKiFormer already achieves strong performance across multiple benchmarks, suggesting that the observed performance improvements from fine-tuning are not solely attributable to the SFT dataset. Subsequently, we conduct controlled SFT comparisons between LoKiFormer and a baseline model (MAP-Neo-7B) using the same SFT dataset. As shown in Table 8, LoKiFormer consistently outperforms the baseline, further indicating that the architectural innovations in LoKiFormer contribute significantly to its performance, rather than any dataset bias. These experiments confirm that the improvements observed in LoKiFormer are primarily due to the model architecture, specifically the integration of Local Fusion Attention (LFA) and the Knowledge Memory Module (KMM), and not to any unintentional bias in the dataset caused by the semantic retrieval filtering process.

## C.3. More Implementation Details

**Hyper-parameters Setting**. All models in the LoKiFormer series (1B, 5B, 7B, 13B, 33B, and 60B) are trained using the AdamW optimizer with momentum parameters $\beta_1 = 0.9$, $\beta_2 = 0.95$, and a weight decay of 0.1 to ensure stable and efficient convergence. The learning rate is carefully tuned for each model scale and training phase. For pre-training, we use an initial learning rate of $8 \times 10^{-3}$ for the 1B model, and $3 \times 10^{-4}$ for the 5B and 7B models, decreasing to $2 \times 10^{-4}$ for the 13B

model, and further to $1 \times 10^{-4}$ for the 33B and 60B models. During supervised fine-tuning (SFT), a lower learning rate is applied: $3 \times 10^{-4}$ for the 1B model, $4 \times 10^{-5}$ for 5B and 7B, $2 \times 10^{-5}$ for 13B, and $1 \times 10^{-5}$ for 33B and 60B. All models are trained using mixed-precision with gradient clipping set to 1.0. The context length during training ranges from 2048 for smaller models to 4096 for the larger variants, as detailed in Section B. All main pre-training experiments use a global batch size of 16,384 and run for a fixed duration of 10,000 steps, consuming hundreds of billions of tokens depending on model context length. For ablation studies, a smaller batch size of 1,024 is used.

**SFT Alignment Strategy.** We emphasize that LoKiFormer models are aligned purely through supervised fine-tuning (SFT) on our curated large-scale instruction dataset described in Section C.2. No reinforcement learning (RL) or preference optimization algorithms (e.g., RLHF, GRPO) are employed at any stage. All alignment behaviors, including instruction following and factual consistency, emerge from supervised training on the cleaned and decontaminated SFT corpus, which contains diverse knowledge-intensive question–answer and multi-turn instruction samples across more than 50 domains. This ensures a fully transparent and reproducible alignment process without reliance on external reward models.

**Details of Parallelism Strategy**. For models scales up to and including 13B parameters, we shard the parametric memory with expert parallelism (EP=2) and otherwise use pure data parallel (DP) replication; pipeline, tensor, and sequence parallelism are disabled. Training runs on 20 nodes with 8 Ascend 910B GPUs each, which meets memory and throughput targets without intra-layer model parallelism. For the 30B and 60B model, we employ an 8 stage pipeline (PP=8) with with expert parallelism of degree four (EP=4) and data parallelism (DP) over the residual dimension; tensor/sequence parallelism are unused. Using Megatron-LM with Transformer Engine, we train stably on H200, validated on a 4-node setup and scaled to a 50-node cluster ($8\times$ H200 per node).

## C.4. More Details of Visualization of Knowledge Field-Domain Associations

To further elucidate the internal mechanisms of the proposed architecture, we conducted a comprehensive visualization analysis (please refer to Figures 2 and 7) to verify whether the Knowledge Block spontaneously acquires structured, domain-specific representations. The primary objective was to observe if high-relevance tokens from distinct domains focus on different key regions via the cross-attention mechanism. Architecturally, a trainable global storage module containing initialized Key-Value pairs ($K_{kb}, V_{kb} \in \mathbb{R}^{N \times d}$, where $N$ denotes the number of knowledge slots) is integrated into the MoE-MLA framework. In this setup, a query vector $Q_{ctx}$ is derived from the intermediate compressed KV representations via a lightweight MLP, facilitating a cross-attention operation defined as:

$$\text{Attention}(Q_{ctx}, K_{kb}, V_{kb}) = \text{softmax}\left(\frac{Q_{ctx}K_{kb}^{\top}}{\sqrt{d}}\right) V_{kb} \tag{15}$$

The resulting output subsequently serves as supplementary input for the MoE experts and the router. For the experimental setup, we utilized multiple sub-domain test sets from the MMLU benchmark, spanning disciplines such as History, Mathematics, Physics, Chemistry, and Politics. For each input sample, we identified high-relevance tokens—such as "Newton," "Calculus," or "Constitution"—using domain dictionary matching and embedding similarity annotation. We then extracted the attention score vectors generated between $Q_{ctx}$ and $K_{kb}$ for these tokens, normalizing them to obtain a probability distribution $\alpha_i \in \mathbb{R}^N$, which represents the focus intensity on specific knowledge slots. The core observations reveal a distinct pattern: high-relevance tokens from different domains exhibit attention scores that are significantly concentrated on disparate subsets of keys. For instance, mathematical tokens predominantly activate specific key indices (e.g., #12, #45, #89), whereas historical tokens focus on a separate set of indices (e.g., #3, #67, #102). Although related fields like Physics and Chemistry show partial overlap, they maintain exclusive high-activation regions. These findings empirically validate that the Knowledge Block possesses the capability to automatically organize domain knowledge into structured partitions without explicit supervision, thereby enabling the model to perform precise, domain-aware retrieval via the query mechanism to distinct knowledge regions.

# D. More Discussions

**Comparisons with Deepseek's Engram** (Cheng et al., 2026). Engram and our LoKiFormer both aim to enhance Transformer models by incorporating externalized knowledge mechanisms, but they differ significantly in their approach to memory handling and retrieval. While Engram employs a static, discrete lookup memory using n-gram embeddings and an O(1) retrieval mechanism, LoKiFormer integrates a fully parametric Knowledge Memory Module (KMM), allowing for end-to-end differentiable learning. Engram's hard lookup mechanism limits adaptability, whereas LoKiFormer's soft retrieval

through dot-product matching enables more dynamic and context-sensitive knowledge access. Additionally, Engram operates as a separate memory system decoupled from computation, relying on parallel memory retrieval, while LoKiFormer integrates its memory module within the Transformer architecture, enhancing flexibility and efficiency in pretraining. LoKiFormer's combination of Local Fusion Attention (LFA) and KMM accelerates convergence, improving both local and global knowledge integration. In summary, Engram excels in static knowledge retrieval, whereas LoKiFormer offers a more adaptable, differentiable approach, providing flexibility in handling dynamic knowledge during training and inference.

**Comparisons with LM2** (Kang et al., 2025). KMM introduces a fixed, parameterized key-value memory structure designed for global knowledge storage and direct retrieval, creating a dedicated knowledge pathway. In contrast, LM2 relies on input-dependent key-value pairs for context modeling, where the KV pairs are dynamically adjusted based on the input. This design allows LM2 to capture context more flexibly but does not provide the same level of structured knowledge retrieval as KMM. The key difference lies in how knowledge is accessed: KMM decouples knowledge storage from computational pathways, enabling more transparent and flexible retrieval, while LM2's memory mechanism is tied to the input sequence, making it less efficient for retrieving global knowledge. In terms of parameter efficiency, KMM introduces only a 2.12% increase in parameters for the 7B model, demonstrating that performance gains come from the novel architecture rather than simple scaling. On the other hand, LM2 typically requires a significant increase in memory capacity, leading to larger parameter budgets. This makes KMM more efficient in terms of both parameter usage and computational cost when compared to LM2 at similar parameter budgets.

**Comparisons with Sliding-window Attention**. While both LFA and sliding-window attention introduce locality, they differ fundamentally in design and purpose: i) No truncation of global context. Sliding-window attention limits each token to a fixed local window by masking, which prevents modeling of long-range dependencies. Instead, our LFA preserves full attention and adds a lightweight convolutional fusion before attention, enriching local representations without restricting the receptive field. ii) Learnable locality rather than fixed windows. In sliding-window attention, the local region is predefined and static. In LFA, the convolutional kernels are learned jointly with the model, allowing flexible adaptation of how local context is fused across layers and heads. iii) Complementary rather than substitutive. Sliding-window attention replaces part of attention computation to save FLOPs, while LFA complements attention by providing locally enhanced inputs. It can thus be seamlessly combined with other attention variants without altering attention sparsity or complexity.

**Throughput and Latency Analysis**. While the convergence speed in the main text is reported in terms of training steps, a complete efficiency evaluation must also consider the computational cost per step. To this end, we compare the training throughput (in tokens per second) of LoKiFormer against the baseline without LFA and KMM under identical hardware configurations ($8\times$NVIDIA A100 GPUs, 80GB memory). The measured throughput for the baseline is 975 tokens/second, while LoKiFormer achieves a throughput of 950 tokens/second. This analysis confirms that the per-step computational overhead introduced by our novel modules is minimal (approximately 2.6%). Therefore, the significant reduction in the number of training steps required to reach the target loss, as reported in the main text, translates directly into a substantial reduction in total wall-clock training time.

We also report the inference efficiency of our LoKiFormer-7B on Ascend 910B (8 NPUs). At 4K context, compared with th baseline, decoding throughput remains nearly unchanged (1299 vs. 1294 tokens/s/NPU, bs=256), while time-to-first-token increases only slightly ($132 \rightarrow 136$ ms). Importantly, LFA does not disrupt the KV cache mechanism. It applies a local transformation before Q/K/V projection, and the resulting K/V are cached as in standard attention. This does not increase KV cache size or introduce additional KV read/write operations.

**More Visualization of Knowledge Field-Domain Associations**. In Figure 7, we present the field-domain associations for the proposed KMM with 64 knowledge fields across layers 8, 12, 16, 20, 24 and 28. These visualizations, generated using the same experimental configuration as the main manuscript, show that individual fields spontaneously evolve to represent domain-specific concepts through end-to-end training. These findings are consistent with the observations in the main text.

**Analysis of Cross-Field Interference via Cosine Similarity.** To directly examine whether different knowledge fields interfere with each other or collapse to similar representations, we analyze the geometry of the learned KMM keys. For the layers visualized in Figures 2 and 7, we take all $F = 64$ key vectors of the knowledge fields and compute the pairwise cosine similarity matrix between field pairs. If multiple fields collapsed or encoded heavily overlapping content, we would expect large off-diagonal cosine values and visible clustered structures in this similarity heatmap.

The resulting cosine-similarity matrix (see Figure 11) is very close to an identity matrix. For instance, in layer 4, the mean off-diagonal cosine similarity is -0.001, and the maximum absolute off-diagonal cosine similarity is only 0.0513. In other

words, different knowledge fields are nearly orthogonal in the learned key space, with no evident clustering or collapse. This indicates that, at $F = 64$, the KMM learns a set of well-separated, specialized fields rather than redundant or interfering ones. While this does not preclude interesting behaviors at much larger scales, it provides concrete empirical evidence that cross-field interference and field collapse are not observed in our current setting.

**Analysis of Local Fusion Operation via Attention Entropy.** To investigate how the proposed Local Fusion Attention (LFA) interacts with the standard self-attention mechanism, we analyze the attention entropy distribution across layers. We argue that LFA offloads the burden of local modeling from the attention mechanism, allowing it to focus on broader, global dependencies. We trained a 1.2B parameter proxy model with 48 layers. LFA modules are inserted with a stride of 4, starting from layer 5. We compare the average attention entropy per layer against a baseline model trained under identical settings.

As shown in Figure 8, 9 and 10, our LoKiFormer exhibits distinct entropy dynamics that validate the "unburdening" hypothesis. First, its early layers (L0-L2) display significantly higher entropy than the baseline, indicating immediate access to a global context, as local processing is offloaded by the architecture's inductive bias. It is important to note that while higher entropy does not directly imply "global understanding," it serves as a relative measure of attention dispersion. Higher entropy in this context reflects more effective aggregation of global information rather than uninformative or unlearned attention. Second, the model reaches peak entropy faster, suggesting more efficient information flow and earlier establishment of global semantics. Crucially, we observe no periodic entropy drops at the LFA insertion points, demonstrating that LFA operates orthogonally to self-attention. Instead of constraining attention periodically, LFA fuses local features into the residual stream, enabling subsequent attention layers to maintain a consistently broad receptive field. Although entropy is analyzed on a 1.2B model for efficiency, the same relative trends between baseline and LFA are observed at 5B, suggesting the effect is not scale-specific. These findings empirically confirm that LFA successfully decouples local feature extraction from global modeling, allowing self-attention to function as a more efficient global aggregator.

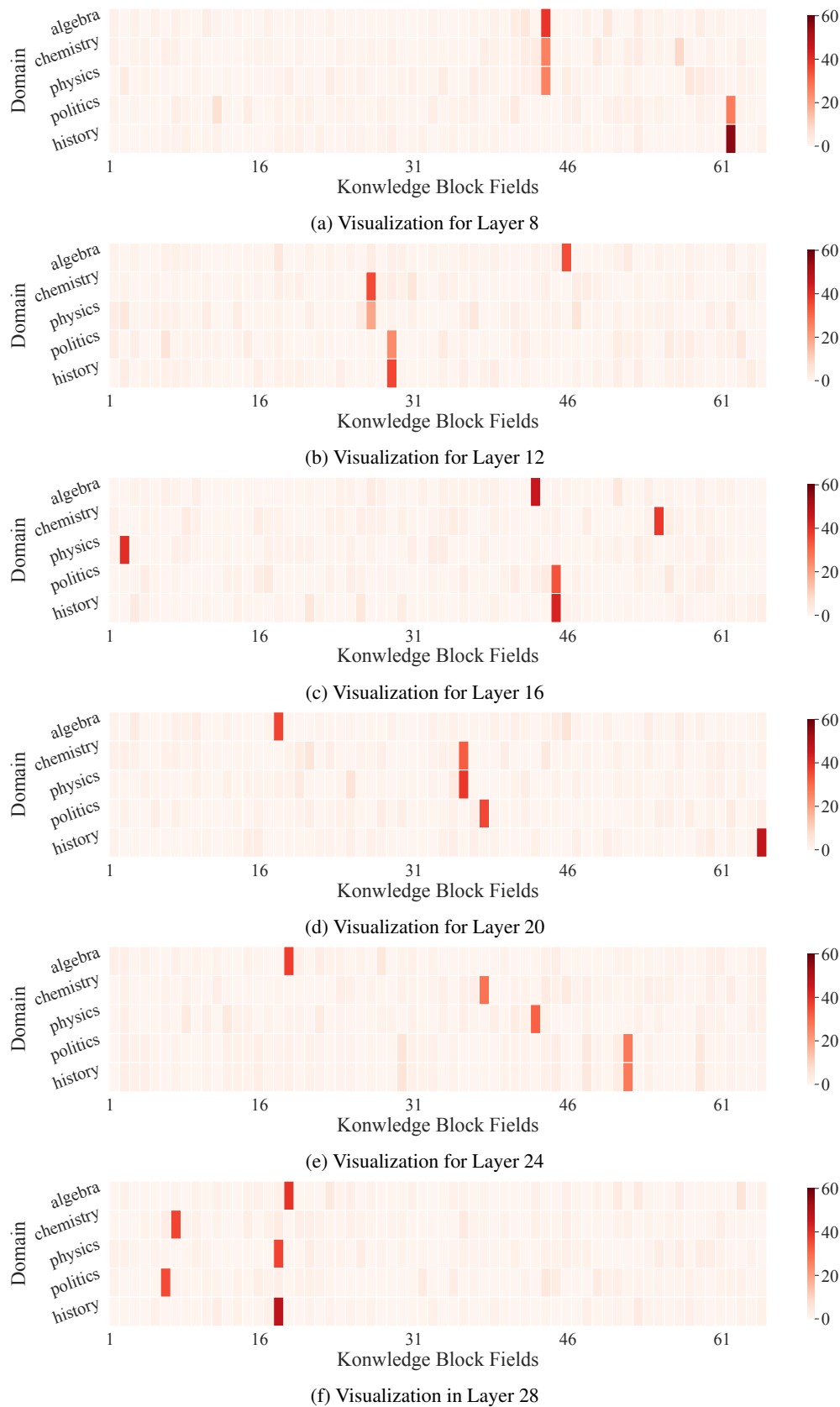

(a) Visualization for Layer 8

(b) Visualization for Layer 12

(c) Visualization for Layer 16

(d) Visualization for Layer 20

(e) Visualization for Layer 24

(f) Visualization in Layer 28

*Figure 7.* Field–domain associations of the proposed KMM with 64 knowledge fields, evaluated on five MMLU domains (1,024 samples each). The heatmap of softmax($\mathbf{H}\mathcal{K}/d_u$) shows that fields emerge with domain-specific specialization.

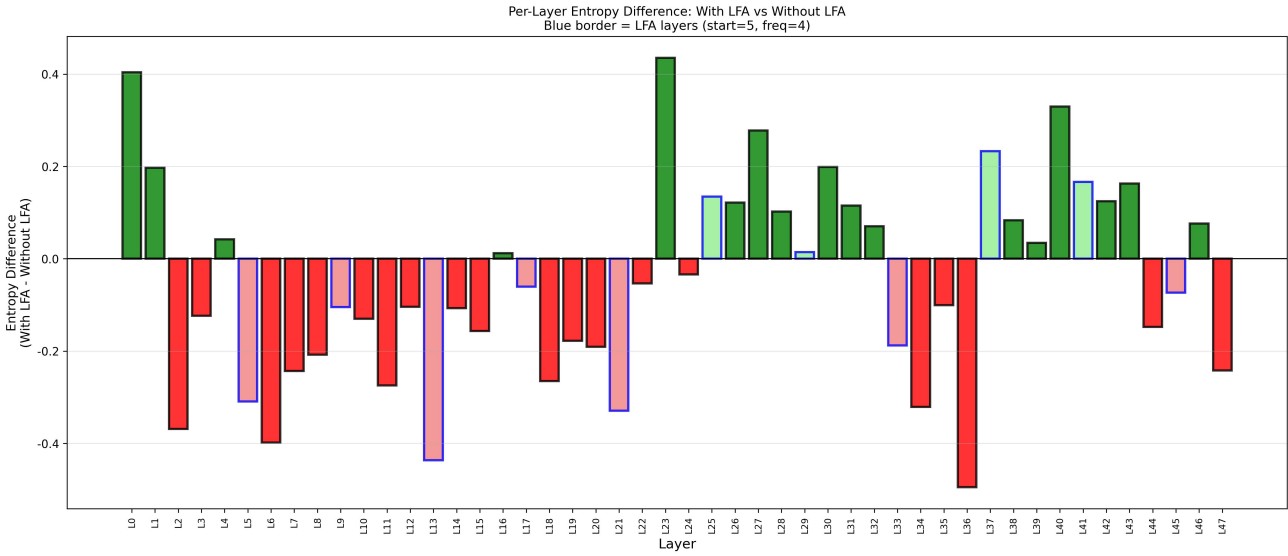

*Figure 8.* Per-layer average attention entropy difference between the model with and without Local Fusion Attention (LFA). Bars with blue borders indicate layers where LFA modules are inserted.

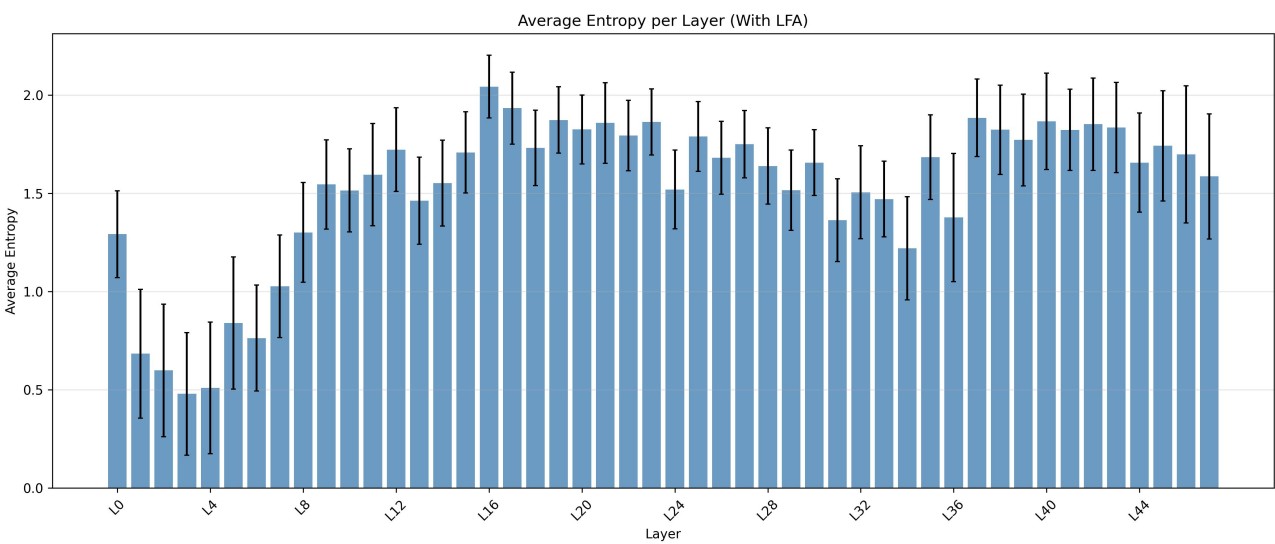

*Figure 9.* Per-layer average attention entropy of the model with Local Fusion Attention (LFA).

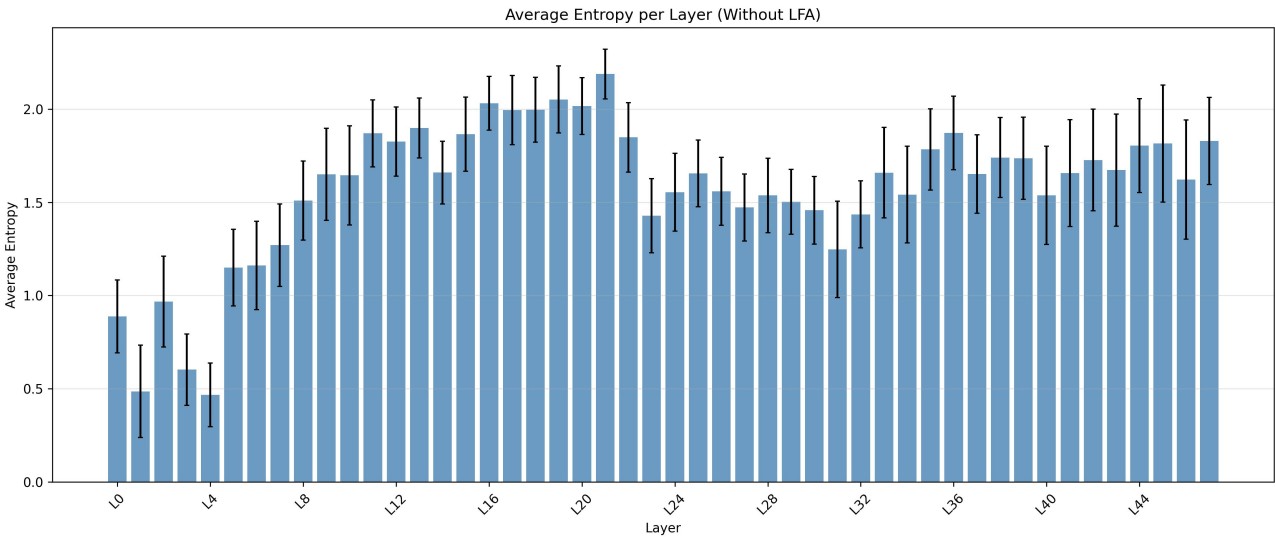

*Figure 10.* Per-layer average attention entropy of the model without Local Fusion Attention (LFA).

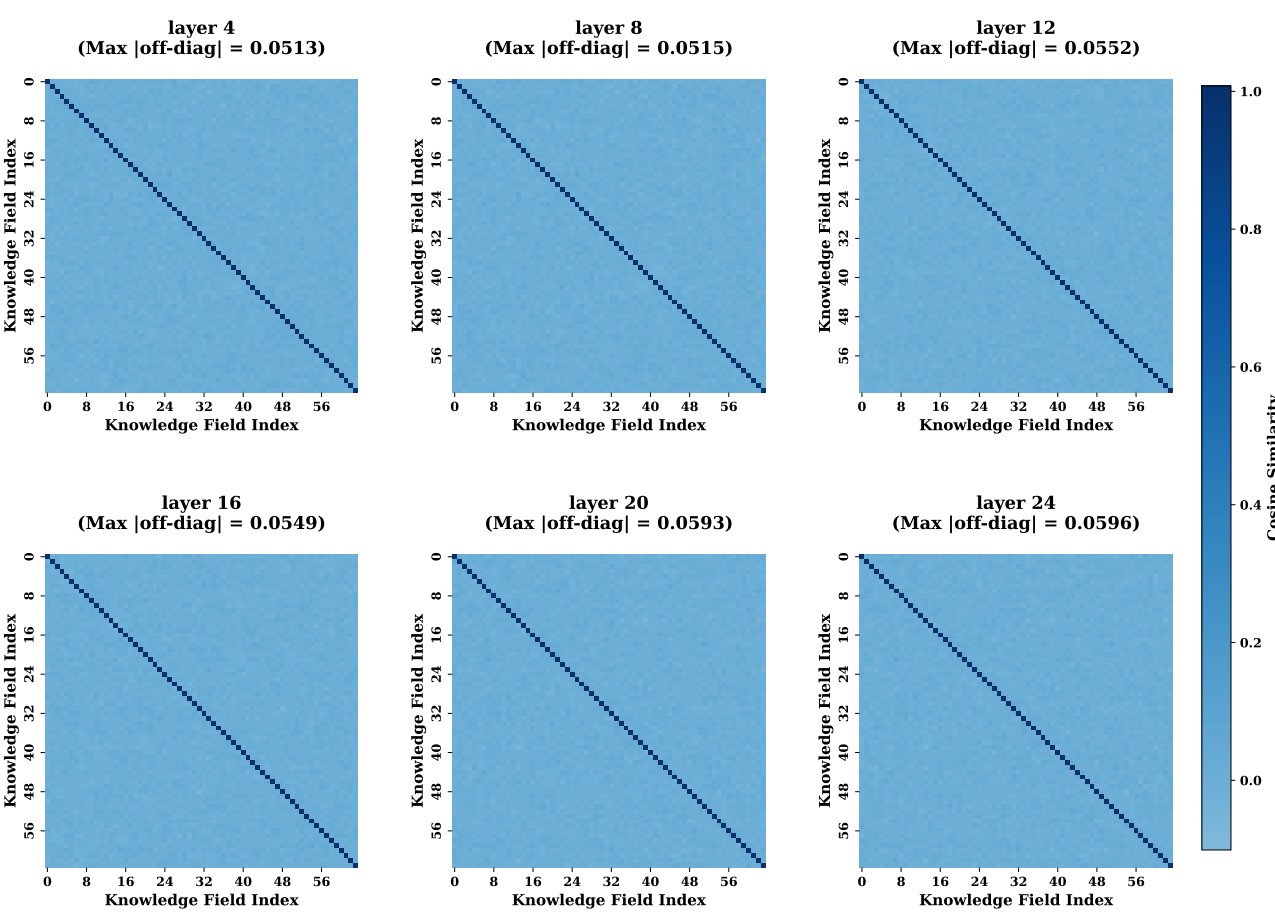

*Figure 11.* Cosine similarity matrix between 64 KMM knowledge fields across different layers.

