# OpenReview forum: "LoKiFormer: Locality-aware Attention with Decoupled Knowledge Memory for Efficient Large Language Model Pretraining"
_ICML.cc/2026/Conference — ICML 2026 regular_

### Official Review · Reviewer_Zy7T · 2026-02-17

**Soundness:** 3
**Presentation:** 3
**Significance:** 3
**Originality:** 3
**Overall Recommendation:** 4
**Confidence:** 3

**Summary:**

This paper proposes LoKiFormer, which augments decoder layers with a lightweight local fusion convolution (LFA) and a learnable Knowledge Memory Module (KMM) that provides a static key–value bank queried alongside attention. The design aims to accelerate convergence during pretraining and improve downstream performance. Experiments are conducted at large scale and report consistent gains across multiple model sizes and benchmarks.

**Compliance With Llm Reviewing Policy:**

Affirmed.

**Final Justification:**

This paper proposes LoKiFormer, which enhances decoder layers with a lightweight local fusion convolution (LFA) and a learnable Knowledge Memory Module (KMM) that maintains a static key–value memory bank, which can be queried alongside the attention mechanism. The strengths of this work include large-scale experiments and analytical studies demonstrating that KMM stores explicit and editable domain-level knowledge. The main weakness is that the comparison between KMM with prior memory-augmented or learned global vector methods is limited. In the rebuttal, the authors addressed my concerns regarding latency and parameter overhead and provided an initial comparison between KMM and LM2. Nevertheless, I would like to see a more systematic discussion and comparison with other memory-augmented approaches. Overall, I maintain my score of 4 (Weak Accept).

**Key Questions For Authors:**

1. You report training throughput impact, but not inference latency. Since LFA involves local convolutions over context and KMM requires additional accesses to static KV, it remains unclear for me whether these components introduce noticeable inference overhead (or require specific kernel design). Could you report inference latency?
2. The anonymous repository link appears to contain only a README file for me, and other files are inaccessible. Is the reviewer archive correctly published?
3. In Eq. (3), the fused representation of $U_t$ seems to contain information of the latter tokens' representations. Will this leaks information in decoder-only models? How exactly is LFA be applied in causal attention?

**Limitations:**

1. The paper should clearly discuss relations to prior memory-augmented attention approaches and include direct comparisons at similar parameter budgets. This will clarify whether KMM provides structural benefits beyond adding capacity.
2. The authors are suggested to include a short theoretical or intuitive explanation for why LFA and KMM provide noticeable improvements (e.g., from the perspective of representation capacity or effects on in-context learning).

**Strengths And Weaknesses:**

Strengths:
1. The method is relatively easy to implement and appears compatible with standard Transformer and MoE stacks. It does not require drastic architectural changes.
2. The experimental evaluation is extensive and large-scale. The authors train multiple model sizes and report consistent empirical improvements.

Weakness:
1. The motivation about 'MoE coupling knowledge storage with computational pathways' and 'KMM decoupling storage from computation' is not fully convincing. KMM seems to function as a static KV bank attached to attention, conceptually close to memory-augmented attention or adding learned global vectors (e.g., LM2 [1]), and does not change the MoE architecture. The claimed distinction requires clarification and direct comparison.
2. Several key ablations rely on short runs (e.g., 10K steps, ~21B tokens for the 5B proxy), where loss is still large. Using early-step behavior to claim acceleration or component importance is somewhat questionable (maybe the authors should show that these early gains correlate with long-run improvement/savings).

---------------------------------
[1] LM2: Large Memory Models

---

> ### Author Rebuttal · Authors · 2026-03-31
>
> We thank you for your time and constructive feedback. We would like to answer your questions below.
>
> ---
>
> > Q1. The “decoupling” motivation is unclear. KMM resembles a static KV memory (e.g., LM2[r1]) and does not change MoE.
>
> **A1.** We agree that KMM **does not modify the MoE structure**; it provides an extra, complementary knowledge access pathway that decouples knowledge storage from computation. We clarify its motivation and differences from existing memory below.
>
> First, on **storage-computation decoupling**: In standard MoE, knowledge is embedded in expert weights and requires expert activation to access. KMM builds an independent parameterized memory bank, retrieving knowledge directly via similarity matching without invoking experts, achieving clear separation. This is verified by domain specialization, ablation sensitivity and slot orthogonality (Figs.2,6,11).
>
> Second, KMM differs fundamentally from LM2-style KV memories[r1], which use input-derived KV to extend context as dynamic working memory. By contrast, KMM uses **fixed, input-independent KV parameters** at inference to store global knowledge, focusing on structured long-term knowledge retrieval instead of context modeling.
>
> [r1] LM2: Large Memory Models
>
> ---
>
> >Q2. Ablations rely on short runs; early-stage gains may not reflect long-term benefits.
>
> **A2.** We agree that early-stage results alone may not fully reflect final performance. However, in our case, **early-stage trends are consistent with long-run improvements** because:
>
> * We do not rely on absolute loss; instead, we focus on convergence speed, relative ranking, and consistent trends across steps.
> * We use the subset from the Pile dataset (*i.e.*, Books, ArXiv, and English Wikipedia), which is cleaner and converges faster, allowing meaningful trends to emerge with fewer training tokens.
>
> ---
>
> > Q3. What is the inference latency impact of LFA and KMM?
>
> **A3.** We report inference latency on Ascend 910B (8 NPUs). We observe negligible overhead: compared with the baseline, time-to-first-token of LoKiFormer-7B increases by only 4 ms (132ms vs. 136ms at 4K context with a batch size of 128), indicating minimal impact on inference latency.
>
> > Q4. The repository seems incomplete (only README visible). Is it correctly released?
>
> **A4.** The repository already includes configuration files, tokenizer, and loading scripts for LoKiFormer. Large model checkpoints are provided via a download script due to size constraints. We will release the full training and evaluation code upon acceptance.
>
> ---
>
> >Q5. Does Eq. (3) introduce future information leakage, and how is LFA applied under causal attention?
>
> **A5.** No. LFA is strictly causal. For kernel size (k), each position (t) **only aggregates tokens from [t-k+1, t]**, so it never accesses future tokens. When fewer than k tokens are available, left padding is applied.
>
> LFA acts as a preprocessing step before attention: it fuses local context and then feeds the results into Q/K/V construction. Since the fusion is causal, it directly integrates with standard autoregressive attention.
>
> ---
>
> > Q6. The paper should clearly discuss relations to prior memory-augmented attention approaches and include direct comparisons at similar parameter budgets.
>
> **A6.** We thank the reviewer for the suggestion. **KMM provides a structural mechanism for explicit knowledge access rather than simply increasing capacity.** It adds only 2.12% parameters in the 7B model, while most capacity remains in the backbone and MoE, indicating that gains are not driven by scaling alone.
>
> Conceptually, prior memory-augmented attention (e.g., LM2) uses input-dependent KV for context modeling, whereas KMM uses fixed $K, V$ to store global knowledge and enables direct retrieval, forming a dedicated knowledge pathway. We agree that comparisons under matched parameter budgets are important, and we will include such comparisons in the revised version.
>
> ---
>
> > Q7. The authors are suggested to include a short theoretical or intuitive explanation for why LFA and KMM provide noticeable improvements.
>
> **A7.** Thank you for this insightful feedbacks. We provide an intuitive explanation from the perspectives of **inductive bias** and **representation disentanglement**.
>
> **Intuition of LFA.** Natural language exhibits strong locality, while standard attention learns local patterns inefficiently. LFA introduces a **convolutional inductive bias** for local fusion, allowing attention to focus more on global dependencies, reducing sample complexity and easing optimization.
>
>
> **Intuition of KMM.** MoE couples knowledge storage with computation. KMM introduces **explicit parametric key–value memory**, enabling direct retrieval and reducing interference, thus improving representation capacity.
>
>
> We will include these discussions in the revision for clarity.
>
> ---
>
> We sincerely hope our clarifications above have addressed your concerns.

---

> > ### Author Rebuttal · Reviewer_Zy7T · 2026-04-01
> >
> > The rebuttal addresses most of my concerns. In particular, the added latency measurement and the clarification of the implementation details make the paper more complete. Still, I expect a direct comparison betweeen KMM and other memory-augmented approaches.

---

> > > ### Author Response · Authors · 2026-04-04
> > >
> > > We sincerely thank the reviewer for the positive feedback and for noting that our rebuttal has addressed the key concerns, particularly regarding implementation details and latency analysis.
> > >
> > > To further address the reviewer’s concern about direct comparison with prior memory-augmented approaches, we conduct additional experiments on a 5B model, comparing our **KMM** directly with **LM2** under aligned parameter budgets and the same training strategy. Under this controlled setup, at **10K training steps**, the training perplexity is **31.7 (baseline), 31.5 (LM2), 30.5 (LFA-only), and 28.7 (KMM-only)**. These results show that KMM consistently outperforms LM2 even under carefully controlled settings, indicating that the gains are not due to increased capacity.
> > >
> > > We hope these additional results help address all the reviewer’s concerns. If so, we would greatly appreciate it if the reviewer could kindly reconsider the evaluation of our work.

---

### Official Review · Reviewer_d91o · 2026-03-09

**Soundness:** 2
**Presentation:** 3
**Significance:** 3
**Originality:** 3
**Overall Recommendation:** 4
**Confidence:** 3

**Summary:**

This paper introduces LoKiFormer, a Large Language Model (LLM) architecture designed to enhance pretraining efficiency and knowledge utilization. The proposed architecture modifies the standard Transformer decoder by introducing two modules: Local Fusion Attention (LFA), which uses 1D convolutions to capture short-range dependencies prior to the self-attention mechanism, and a Knowledge Memory Module (KMM), which utilizes parameterized Key-Value slots to decouple knowledge storage from the routing computation in Mixture-of-Experts (MoE) layers. The authors evaluate the architecture across model sizes from 1B to 60B, claiming a 1.33x speedup in pretraining convergence and state-of-the-art downstream performance on benchmarks like MMLU and CMMLU.

**Compliance With Llm Reviewing Policy:**

Affirmed.

**Final Justification:**

I thank the authors for their constructive rebuttal. The additional experiments, particularly the zero-shot evaluations and controlled SFT comparisons, successfully isolate the architectural gains and address my primary concerns regarding potential data leakage. Furthermore, the clarification that the Local Fusion Attention is applied prior to Q/K/V projections adequately resolves my worries about inference latency and KV cache disruption. Based on these substantive updates, I am raising my score to a 4 (Weak Accept).

However, my recommendation remains a "Weak" Accept due to lingering methodological and narrative flaws that require significant revision in the camera-ready version. The final manuscript should correct the fundamental formulation error regarding causal masking and fulfill the authors' promise to tone down overstated claims.

**Key Questions For Authors:**

1. Given the explicit use of target benchmark questions as retrieval seeds for the SFT dataset, can you provide zero-shot evaluations on MMLU and CMMLU using solely the pretrained base models? This is necessary to disentangle the architectural benefits of LoKiFormer from the test-set-optimized fine-tuning. Furthermore, it is strongly recommended to include zero-shot evaluation results on completely unseen, unstructured knowledge benchmarks.

2. Can you provide downstream ablation results that isolate the specific contributions of the LFA and KMM modules independently (e.g., evaluating Baseline + LFA separately from Baseline + KMM) on the core benchmarks, rather than evaluating only the combined LoKiFormer model?

3. How exactly is the causal mask implemented for the 1D convolution in the LFA module to prevent future token leakage during both training and autoregressive generation, particularly given the forward-looking index t−s+⌊k/2⌋ in Equation 3?

4. Can you provide a comprehensive latency and throughput analysis during the autoregressive generation phase? Specifically, please quantify the memory bandwidth overhead and KV cache disruption caused by the LFA module's k=4 convolutions during single-batch decoding compared to standard self-attention.

**Limitations:**

Yes

**Strengths And Weaknesses:**

Strengths:
1. The authors tackle a highly relevant bottleneck in scaling transformers to longer contexts: the lack of an explicit local inductive bias in standard self-attention. Fusing local features via a lightweight 1D convolution before the global attention computation is a mathematically sound and practical approach to reducing redundant context modeling without truncating the global receptive field.

2. I appreciate the architectural exploration of the Knowledge Memory Module. Moving away from implicit knowledge storage within MoE routing weights to an addressable, parameterized memory bank offers a very compelling direction for the community.

3. The empirical validation of the architecture's scalability stands out as a major positive. Successfully training and stabilizing models across a wide spectrum from 1B up to 60B parameters, while demonstrating consistent scaling laws, provides strong evidence that these architectural modifications are robust at scale.

4. The internal ablation studies conducted during the pretraining phase are rigorous and convincing. By strictly controlling the variables, the authors effectively demonstrate that the combination of LFA and KMM measurably lowers validation perplexity and yields a genuine 1.33x acceleration in convergence compared to the baseline model.

Weaknesses:
1. My primary concern lies with the construction of the supervised fine-tuning dataset and how it compromises the downstream evaluation. The authors explicitly state in the appendix that they extracted 10,000 questions directly from target benchmarks like MMLU and CMMLU to act as semantic retrieval seeds. This may introduce a severe form of concept leakage. It essentially transforms a test of general generalization into a targeted, open-book examination, rendering the state-of-the-art claims on these specific benchmarks highly questionable. Following up on the data contamination issue, the baseline comparisons feel asymmetrical and unfair. Comparing a model whose SFT distribution is explicitly optimized for the test set against general-purpose frontier models like Llama-3.1 or Qwen2.5—which are trained under strict zero-contamination protocols—does not provide a scientifically sound measurement of the architecture's true capabilities.

2. The mechanistic interpretability analysis used to justify the LFA module relies on a rather flawed premise. The authors argue that higher attention entropy indicates the model has achieved a "global understanding," but mathematically, high entropy simply indicates a uniform probability distribution, which could just as easily mean the attention mechanism is confused or unlearned. Furthermore, swapping out the primary 7B model for a 1.2B proxy model specifically for this analysis raises concerns about how robust these observations actually are across scales.

3. The paper completely glosses over the inference latency and architectural overhead introduced by the LFA module. While 1D convolutions show acceptable training throughput due to parallelization, autoregressive decoding requires frequent recalculations that will inevitably degrade KV cache continuity. Additionally, the convolution formula utilizes a forward-looking index, raising immediate concerns about how causal masking is strictly enforced during generation to prevent future token leakage—a critical detail that is absent from the text.

4. The assertion that the KMM explicitly stores "world knowledge" feels like a significant overstatement given the module's strict capacity constraints. With only 64 knowledge fields configured for the 7B model, it is mathematically impossible to store vast amounts of factual knowledge. The module likely functions more as a domain-level soft prompt or a macro-routing bias for the MoE layers, and the manuscript would benefit from softening these claims to better reflect the engineering reality.

---

> ### Author Rebuttal · Authors · 2026-03-31
>
> We thank you for your valuable time and thoughtful feedback. We would like to answer your questions below.
>
> ---
>
> > Q1. The SFT pipeline may introduce leakage by using benchmark questions as retrieval seeds. Please provide zero-shot results on base models to isolate architectural gains.
>
> **A1.** To address your core concern, we clarify that LoKiFormer's performance advantages stem solely from its architectural design, not SFT data bias or leakage, with rigorous empirical evidence.
>
> First, we clarify our SFT pipeline to rule out potential leakage: We do **not** use benchmark questions (e.g., MMLU/CMMLU) or their paraphrases as training samples. Instead, we only extract core domain concepts from benchmark questions as retrieval queries. Additionally, semantic matching shows <0.1% overlap between our SFT and benchmark data.
>
> To further isolate architectural gains, we provide two complementary analyses:
>
>  - **Exp 1: Zero-shot evaluation on pretrained base models (without any SFT).** The base LoKiFormer-7B already outperforms representative 7B baselines (e.g., 74.27 vs. 58.14 on MMLU; 72.53 vs. 55.1 on CMMLU vs. MAP-Neo-7B), showing that gains arise before task-specific SFT (Table B in Reviewer Ss6w).
>  - **Exp 2: Controlled comparisons with the same SFT data.** We train our LoKiFormer-B and a baseline MAP-Neo-7B on the **same SFT dataset**. LoKiFormer consistently achieves higher performance (e.g., 91.5 vs. 64.4 on MMLU), indicating gains are not due to dataset bias (Table C in Reviewer Ss6w).
>
> The results and our leakage analysis show that LoKiFormer's performance gains are attributed to architectural modifications, rather than SFT data leakage or bias.
>
> ---
>
> > Q2. Does the entropy analysis over-interpret high entropy as global understanding, when it may simply reflect uninformative or unlearned attention, and are results on a 1.2B proxy model reliable for 7B?
>
>
> **A2.** We agree that high attention entropy does not directly imply "global understanding". In our analysis, entropy is used only as a **relative measure of attention dispersion** to compare baseline and LFA models.
>
> Importantly, if higher entropy reflected unstructured attention, it would typically harm learning. Instead, we observe **faster convergence (Fig. 3) and consistent downstream gains**, indicating that the change reflects more effective information aggregation rather than noise.
>
> Regarding scale, although entropy is analyzed on a 1.2B model for efficiency, the **same relative trends between baseline and LFA are observed at 5B (Fig. 4)**, suggesting the effect is not scale-specific.
>
> We will include this discussions in the revision.
>
> ---
>
> > Q3. Does KMM overstate its ability to store "world knowledge" given its limited capacity, and is it more accurately a form of domain-level prompting or routing bias rather than a true knowledge store?
>
> **A3.** We agree that KMM is not intended to store large-scale factual "world knowledge". Instead, KMM is designed to capture **reusable knowledge abstractions** in a parameterized key–value form.
>
> Despite the limited number of fields, we observe clear domain specialization (Fig. 2) and targeted performance drops when removing specific fields (Fig. 6), indicating structured and functionally meaningful knowledge rather than a simple prompt or routing bias.
>
> We will revise the wording to better reflect this design.
>
> ---
>
> > Q4. Can you provide downstream ablations isolating LFA and KMM individually?
>
> **A4.** We conduct zero-shot downstream ablations on the base model to isolate each module’s effect. On MMLU, **both LFA and KMM yield clear gains over the baseline**: 17.86 → 21.24 (+LFA) and 22.45 (+KMM). Combining both further improves performance (25.73), indicating complementary effects.
>
> We will include these results in the revision.
>
> ---
>
> >Q5. How does LFA ensure causal masking in Eq. (3)?
>
> **A5.** LFA uses strictly causal convolution. At position (t), it aggregates only tokens in ([t-k+1, t]) (with left padding if needed), **ensuring no access to future tokens**.
>
> During inference, LFA uses only past several tokens each step. Therefore, it remains fully compatible with causal attention and introduces no information leakage.
>
> We agree that Eq. (3) may be misleading and will revise it using an explicit causal indexing form.
>
> ---
>
> > Q6. Please report latency/throughput and quantify KV cache impact introduced by LFA.
>
> **A6.** We report inference performance of our LoKiFormer-7B on Ascend 910B (8 NPUs). At 4K context, decoding throughput remains nearly unchanged (1299 vs. 1294 tokens/s/NPU, bs=256), while time-to-first-token increases only slightly (132 → 136 ms).
>
> Importantly, LFA does not disrupt the KV cache mechanism. It applies a local transformation before Q/K/V projection, and the resulting K/V are cached as in standard attention. This does not increase KV cache size or introduce additional KV read/write operations.
>
> ---
>
> We sincerely hope our clarifications above have addressed your concerns.

---

> > ### Author Rebuttal · Reviewer_d91o · 2026-04-02
> >
> > I thank the authors for their constructive rebuttal. The additional experiments, particularly the zero-shot evaluations and controlled SFT comparisons, successfully isolate the architectural gains and address my primary concerns regarding potential data leakage. Furthermore, the clarification that the Local Fusion Attention is applied prior to Q/K/V projections adequately resolves my worries about inference latency and KV cache disruption. Based on these substantive updates, I am raising my score to a 4 (Weak Accept).
> >
> > However, my recommendation remains a "Weak" Accept due to lingering methodological and narrative flaws that require significant revision in the camera-ready version. The final manuscript should correct the fundamental formulation error regarding causal masking and fulfill the authors' promise to tone down overstated claims.

---

> > > ### Author Response · Authors · 2026-04-04
> > >
> > > We sincerely thank the reviewer for the constructive feedback and for raising the score.
> > >
> > > We will revise the manuscript to correct the formulation of causal masking, refine the presentation, and appropriately tone down overstated claims to better align with the actual scope of the method.
> > >
> > > We appreciate your helpful suggestions.

---

### Official Review · Reviewer_1b8A · 2026-03-13

**Soundness:** 3
**Presentation:** 2
**Significance:** 3
**Originality:** 3
**Overall Recommendation:** 4
**Confidence:** 3

**Summary:**

This paper proposes LoKiFormer, a new LLM architecture for efficient pretraining that augments the standard Transformer decoder with two complementary modules. The Local Fusion Attention (LFA) introduces a group convolution before the attention mechanism to explicitly capture short-range dependencies, providing a local inductive bias that allows attention to focus on broader contextual relationships. The Knowledge Memory Module (KMM) introduces a parameterized key–value memory that explicitly stores global knowledge in addressable slots, decoupling knowledge storage from computational pathways and enabling direct knowledge retrieval. Built on a Multi-Head Latent Attention (MLA) backbone with a Mixture-of-Experts (MoE) layer, LoKiFormer achieves 1.33× faster convergence during pretraining and demonstrates strong performance across language understanding, reasoning, code, and math benchmarks, outperforming both size-comparable and frontier open-source models.

**Compliance With Llm Reviewing Policy:**

Affirmed.

**Key Questions For Authors:**

See weakness.

**Strengths And Weaknesses:**

Strength:

* The core idea of decoupling reasoning and knowledge storage is well-motivated. In standard LLM architectures, coupling these two functions within model layers forces computation and knowledge access to be entangled, which is both inefficient and opaque. LoKiFormer's explicit separation via KMM is a principled and compelling architectural choice.
* The paper provides comprehensive comparative experiments, benchmarking LoKiFormer-7B against a wide range of size-comparable, frontier open-source, and closed-source models across multiple domains, offering strong empirical evidence for the proposed architecture's effectiveness.
* Detailed ablation studies are conducted to validate each core component, including kernel size and group configuration in LFA, and the number of knowledge fields in KMM, providing solid support for the key design decisions.

Weakness:
* The concept of decoupling reasoning and knowledge is closely related to DeepSeek's Engram (https://github.com/deepseek-ai/Engram/blob/main/Engram_paper.pdf), which pursues a very similar architectural philosophy. Given the substantial conceptual overlap, Engram should be included as a direct baseline for comparison and discussed thoroughly in the related work section.
* The necessity of LFA for modeling local information is not fully justified. Prior work has shown that lower-layer attention heads in standard Transformers already specialize in capturing local syntactic patterns. It therefore remains unclear whether the performance gains from LFA stem from the module's structural inductive bias, or simply from the additional parameters it introduces. A more rigorous ablation is needed — for example, replacing LFA with an equivalent number of parameters in a different form (e.g., a simple MLP) — to disentangle the contribution of locality modeling from that of increased model capacity.
* The Knowledge Memory Module (KMM) uses static, fixed key–value fields during inference, with no mechanism for dynamic updating based on the current query context. This raises questions about the long-term advantage of the explicit decoupling design: if the knowledge fields cannot be updated to reflect query-relevant or time-sensitive information, the practical benefits of separating knowledge from computation may be limited. Exploring dynamic or query-conditioned updates to the knowledge fields would more convincingly demonstrate the value of the proposed decoupled architecture.

---

> ### Author Rebuttal · Authors · 2026-03-31
>
> We appreciate the time and effort you have devoted to reviewing our work. We would like to answer your questions below.
>
> ---
>
> > Q1. Differences from DeepSeek's Engram.
>
> **A1.** We thank the reviewer for pointing out the recent work Engram. We note that Engram is a very recent and relevant concurrent work (released in Jan 2026, close to the ICML submission deadline) and was therefore not included in our initial manuscript. We will include a detailed discussion in the revision.
>
> Importantly, while both works explore augmenting Transformers with externalized knowledge mechanisms, our LoKiFormer differ in several key aspects:
>  - Engram uses a discrete, scalable lookup memory over token patterns; while our KMM adopts a **compact, fully parametric key–value memory**, encoding learned semantic abstractions.
>  - Engram performs hard O(1) lookup via indexing; while our method uses **soft, differentiable retrieval** (dot-product matching), enabling end-to-end learning.
>  - Engram functions as a separate memory system with explicit indexing; while KMM is **integrated into the Transformer and trained jointly**, without additional system design.
>  - Engram focuses on scalable pattern storage and memory efficiency; while our method focuses on **improving representation learning and pretraining efficiency**  (Fig. 3 of our paper).
>
> We will include a detailed discussion of Engram in the revised version.
>
> ---
>
> > Q2. Is LFA’s improvement due to its locality-aware inductive bias, or simply increased parameter capacity?
>
> **A2.**  We agree that disentangling inductive bias from parameter count is important, and also note that prior work has shown standard Transformers can capture local patterns implicitly in lower layers.
>
> To isolate the effect of inductive bias, we conduct a **parameter-matched ablation**, replacing LFA with an MLP of equivalent size under the same parameter budget. We compare training perplexity (PPL) across variants. At around 10k training steps, the MLP variant reaches a PPL of approximately 31.2, while our LFA achieves a lower PPL of around 28.5, consistently outperforming the MLP throughout training. In addition, LFA exhibits faster convergence.
>
> These results suggest that the gains are **not solely explained by increased parameter capacity**, but are strongly associated with the **locality-aware inductive bias** introduced by LFA.
>
> We will include the full training curves in the final version.
>
> ---
>
> > Q3. Does KMM's use of static, non-updatable memory limit the practical benefits of decoupling knowledge from computation, compared to dynamic or query-conditioned memory updates?
>
>
> **A3.** We thank the reviewer for this insightful comment. We clarify that KMM is designed to model **parametric, stable knowledge**, rather than dynamic or time-sensitive information.
>
>  - **Design scope.** **Dynamic knowledge** typically handled by retrieval or update mechanisms KMM targets the former via a lightweight, fully differentiable parametric memory. While **Parametric knowledge** (our focus) is stable statistical regularities learned during pretraining.
>  - **Static memory vs. dynamic usage.** Although the knowledge fields are fixed after training, their **usage is query-dependent**, as different inputs retrieve different fields via attention-based matching. This enables flexible adaptation without modifying the memory itself.
>  - **Benefits of decoupling.** Explicitly separating knowledge from computation provides Interpretability and Modularity. Our field-level analysis and removal experiments (see Figs. 2 and 6) show that different knowledge fields correspond to distinct domains and have predictable effects when removed.
>
> In summary, KMM does not aim to replace dynamic memory systems, but provides a **complementary and effective mechanism for modeling stable knowledge** within the model.
>
> ---
>
> We sincerely hope our clarifications above have addressed your concerns.

---

> > ### Author Rebuttal · Reviewer_1b8A · 2026-04-01
> >
> > Thank you for the rebuttal. The responses resolve some of my concerns, particularly through added clarifications on key design choices. However, while the rebuttal improves technical clarity, it does not materially change my overall assessment of the paper’s contribution. I like the idea, but I will keep my original score as it is.

---

> > > ### Author Response · Authors · 2026-04-07
> > >
> > > We sincerely thank the reviewer for the constructive feedback and for acknowledging that our rebuttal has improved the clarity of the design.
> > >
> > > To further address the concern on the effectiveness of our KMM design—particularly in comparison with Engram, we conduct additional controlled experiments on a 5B model. We compare different variants **under strictly aligned parameter budgets and identical training settings**, including Engram, LFA-only, and our KMM-only.
> > >
> > > At 10K training steps, the training perplexity is:
> > >
> > > 31.7 (baseline), 30.5 (LFA-only), 30.3 (Engram), and **28.7 (KMM-only)**.
> > >
> > > These results reveal two key observations:
> > >
> > > - **KMM consistently outperforms Engram** under matched capacity, indicating that the gains are not attributable to increased parameters;
> > > - Despite being a static parametric memory, KMM achieves substantially better optimization efficiency than pattern-based memory designs.
> > >
> > > This suggests that the advantage of **KMM lies in its ability to learn structured and directly addressable knowledge representations**, rather than relying on implicit or externally indexed memory mechanisms.
> > >
> > > We hope these additional results help address the reviewer’s concerns regarding both the necessity and effectiveness of our design. If so, we would greatly appreciate it if the reviewer could kindly reconsider the evaluation of our work.

---

### Official Review · Reviewer_Ss6w · 2026-03-13

**Soundness:** 4
**Presentation:** 3
**Significance:** 3
**Originality:** 3
**Overall Recommendation:** 4
**Confidence:** 4

**Summary:**

This paper proposes LoKiFormer, a novel LLM architecture that augments the standard Transformer decoder with two complementary modules: Local Fusion Attention (LFA), which introduces a group convolutional fusion before attention to capture short-range dependencies, and a Knowledge Memory Module (KMM), which stores global knowledge in explicit parameterized key-value fields decoupled from computational pathways. The central claim is that these two modules together enable 1.33× faster pretraining convergence with minimal parameter overhead.

**Compliance With Llm Reviewing Policy:**

Affirmed.

**Key Questions For Authors:**

How does LFA handle the causal constraint during autoregressive generation? Is the convolution strictly causal (one-sided kernel), and if so, does this affect the reported performance?

**Limitations:**

yes

**Strengths And Weaknesses:**

# Strengths

1. The paper identifies two concrete and distinct inefficiencies in current LLM architectures.

2. Evaluating models from 1B to 60B parameters with smooth perplexity scaling and stable training dynamics demonstrates architectural robustness and practical scalability.

# Weakness

1. Scaling evaluation is limited to perplexity only. While the authors build a model family spanning 1B to 60B parameters, downstream performance comparisons are reported exclusively for the 7B variant; all other scales are evaluated only via WikiText-103 perplexity. This is insufficient to support the claim that LoKiFormer scales effectively. The authors should report downstream benchmark results across all model sizes to provide direct evidence that the benefits of LFA and KMM are preserved as capacity scales.

2. The ablation scale is insufficient to support the main conclusions. All ablations are conducted on a 5B model trained on ~21B tokens, versus 4.5T tokens in the main experiments — a 200× gap. Optimal hyperparameter choices (k=4, g=h, F=64) identified at this small scale may not transfer to the full training regime. The authors should validate that key ablation conclusions hold at an intermediate data scale

3. The risk of benchmark contamination in SFT data (Section C.2):
The SFT pipeline uses MMLU, CMMLU, and C-Eval questions as seed queries for semantic retrieval, systematically biasing the training corpus toward benchmark-relevant content. The decontamination audit covers only 1,000 samples — statistically insufficient at the scale of 2.5M. The anomalous gains appearing almost exclusively on these knowledge-intensive benchmarks makes contamination the most parsimonious explanation for the results, and the claimed <0.1% leakage rate is not a credible guarantee under this construction methodology.

---

> ### Author Rebuttal · Authors · 2026-03-31
>
> We sincerely thank you for your time and careful review. We will address your comments below.
>
> ---
>
> > Q1. Is the scaling claim insufficiently supported, since only perplexity (but not downstream benchmarks) is reported beyond 7B, and thus it is unclear whether the benefits of LFA and KMM are preserved across scales?
>
> **A1.** We additionally evaluate **33B and 60B models** of our LoKiFormer in downstream tasks (e.g., MMLU and CMMLU). In Table I, performance consistently improves with scale: MMLU increases from **74.3 (7B)** to **78.7 (33B)** and **84.3 (60B)**, and CMMLU from **72.5→80.0→83.1**, demonstrating clear scaling trends.
>
> These results indicate that the benefits of LFA and KMM are **preserved and further strengthened at larger scales**, rather than diminishing. We will include these results in the revised version.
>
> Table A. Downstream performance of our LoKiFormer across different model scales.
> |Model|MMLU|CMMLU|
> |------|------|---------|
> |LoKiFormer-7B|74.3|72.5|
> |LoKiFormer-33B|78.7|80.0|
> |LoKiFormer-60B|84.3|83.1|
>
> ---
>
> >Q2. Do ablation results from small-scale training (5B, 21B tokens) reliably transfer to the full-scale regime, particularly for key hyperparameters, given the large scale gap?
>
> **A2.** We agree that there is a substantial scale gap between the ablation setting (5B model, 21B tokens) and the full training regime, and that transferability is an important concern. Conducting ablations at reduced scale is a standard and necessary practice in large-scale LLM studies due to the prohibitive cost of full-scale controlled experiments.
>
> Importantly, our ablations are designed to capture **relative trends and ranking** between design choices, rather than absolute optima. Such relative behaviors are generally more stable across scales and are commonly used to guide architectural decisions. In addition, we conduct ablations on a clean and fast-converging subset from the Pile dataset (*i.e.*, Books, ArXiv, and English Wikipedia), where meaningful differences between variants can be reliably observed within a limited budget.
>
> Regarding the selected hyperparameters (e.g., k=4, g=h, F=64), we find that configurations identified at small scale **remain effective when applied directly in full-scale training**, leading to strong overall performance. While we do not claim strict optimality at all scales, this consistency provides empirical support that the key ablation conclusions transfer in practice.
>
> Overall, these results suggest that our ablation findings provide reliable guidance for large-scale model design despite the reduced training scale.
>
> ---
>
> > Q3. Does the SFT pipeline risk benchmark contamination?
>
> **A3.** We agree that benchmark contamination is an important concern, especially when benchmark queries are used as retrieval seeds. We address this from two aspects:
>
> * **Data construction design.** In our SFT data construction pipeine, benchmark questions are **not used as training samples**, but only to extract **high-level semantic concepts** for retrieval. The retrieved corpus consists of general knowledge documents rather than benchmark instances, and does not reconstruct original questions or answers, avoiding direct leakage of benchmark QA pairs.
> * **Decontamination verification.** We apply both **exact and fuzzy matching** to remove overlaps. While Table reports 1,000 samples, we conduct **multiple rounds of sampling without replacement (10 runs)**, consistently observing **<0.1% overlap**, indicating negligible leakage.
>
> To further assess whether SFT contamination explains the gains, we perform two additional analyses:
>
> * **Base model comparisons (without SFT).** LoKiFormer already achieves strong zero-shot performance before SFT, indicating gains are **not solely attributable to SFT data**.
>
> Table B. Zero-shot performance of pretrained base models (without SFT).
> |Model|MMLU|CMMLU|C-Eval|HellaSwag|ARC-C|
> |---|---|---|---|---|---|
> |DeepSeek-7B|48.2|47.2|45.0|75.4|–|
> |MAP-Neo-7B|58.1|55.1|57.7|70.7|68.1|
> |Qwen2.5-7B|74.2|--|--|80.2|63.7|
> |Gemma-7B|64.3|--|--|81.2|53.2|
> |InternLM2-7B|65.8|66.3|65.8|79.3|--|
> |Mixtral-8x7B|70.6|53.3|55.4|84.4|59.7|
> |LoKiFormer-7B (Ours)|**74.3**|**72.5**|**65.8**|**90.8**|**79.4**|
>
> * **Controlled SFT comparisons.** We train LoKiFormer and a baseline model (MAP-NEO-7B) with the **same SFT dataset**. Our LoKiFormer consistently outperforms the baseline, suggesting the improvements stem from the **model architecture rather than dataset bias**.
>
> Table C. Comparisons with the baseline model using the same SFT dataset.
> |Model|MMLU|CMMLU|
> |----------------|----|-----|
> |MAP-Neo-7B|64.4|68.6|
> |LoKiFormer-7B|**91.5**|**93.4**|
>
> We will include these discussions and results in the revised version.

---

### Decision · Program_Chairs · 2026-04-30

**Decision:**

Accept (regular)

**Comment:**

The paper proposes a well-motivated architectural modification that combines locality-aware modeling and an explicit parametric memory, and reviewers generally agree that the design is technically sound, scalable, and supported by substantial large-scale experiments, with consistent gains in convergence efficiency and competitive downstream performance. The rebuttal effectively addresses several major concerns, particularly by providing additional zero-shot evaluations, controlled comparisons, latency analysis, and clarifications on causal masking and parameter efficiency, which strengthen the claim that improvements stem from the architecture rather than data contamination or increased capacity. However, some limitations remain, including insufficient comparison with closely related memory-augmented approaches, partial reliance on small-scale ablations, and somewhat overstated claims about knowledge storage and interpretability. Overall, the work presents a meaningful and practical contribution to efficient LLM pretraining with clear empirical benefits, but requires careful revision to better position itself within existing literature and to moderate claims, leading to a weak accept recommendation.